



# Fire and vegetation dynamics in North-West Siberia during the last 60 years based on high-resolution remote sensing

Oleg Sizov[1,*], Ekaterina Ezhova[2,*], Petr Tsymbarovich[3], Andrey Soromotin[4], Nikolay Prihod'ko[4],
Tuukka Petäjä[2,4], Sergej Zilitinkevich[2,5], Markku Kulmala[2,4], Jaana Bäck[6], and Kajar Köster[6]

[1]Institute of Oil and Gas Problems Russian Academy of Science, Moscow, Russia
[2]Institute for Atmospheric and Earth System Research (INAR)/Physics, University of Helsinki, Finland
[3]Institute of Geography Russian Academy of Science, Moscow, Russia
[4]Tyumen' State University, Tyumen', Russia
[5]Finnish Meteorological Institute, Helsinki, Finland
[6]Institute for Atmospheric and Earth System Research (INAR)/Forest Sciences, University of Helsinki, Finland
[*]These authors contributed equally to this work.

**Correspondence:** Ekaterina Ezhova (ekaterina.ezhova@helsinki.fi)

**Abstract.** Rapidly warming Arctic undergoes transitions that can influence global carbon balance. One of the key processes is the shift towards plant species with higher biomass underlining a stronger carbon sink. The shift is predicted by the models based on abiotic climatic factors but it is not always confirmed with observations. Here we use high-resolution remote sensing to study the process of transition of tundra into forest on the 20 000 km$^2$ area in North-West Siberia. Overall, 40% of the study

area was burned during 60-yr period. Three quarters of the burned areas were dry tundra. Ca 10% of the study area experienced 2-3 fires with an interval of 15-60 years, suggesting a shorter fire return interval than that reported earlier for the northern areas of Central Siberia (130-350 years). Based on our results, the shift in vegetation (within the 60-years period) occurred in 40-85% of the territories that experienced fires, suggesting a strong role of disturbances for the tree advance. All fire-affected territories were flat, therefore no effect of topography was detected. Oppositely, in the undisturbed areas, tundra-forest transition was

observed only in 6-15% of the territories, characterized by a steeper topographic slope. Our results show that the fires often originated near the centres of anthropogenic activity, which is continuously increasing due to the economic importance of the region. This might explain larger frequency of major fires in the northern territories of West Siberia compared to Central Siberia.

## 1 Introduction

North-West Siberia is the region subject to a strong warming trend in summer as compared to the Arctic average. The annual warming trend reported for the entire Arctic (data set for 1971-2017) is 0.6°C per decade, resulting from the cold season trend of 0.7°C per decade, and the warm season (June-September) trend of 0.4°C per decade (Box et al., 2019) in the Arctic as a whole. According to the 2nd Assessment report on the climate change on the Russian territory (Katsov et al., 2014), the winter warming trend in North-West Siberia (data set from 1972–2012) is 0.4-0.7°C per decade, which is comparable to the

trends reported for the entire Arctic. However, the summer trend is 0.8-1.0°C higher per decade, double that is reported for



the entire Arctic. At the same time, meteorological observations indicate that snow cover thickness increased at the rate 2-10 cm per decade but the number of snow cover days decreased at the rate up to 8 days per decade (Katsov et al., 2014). An increase in warm degree-days favors a shift of vegetation type towards more southern species, and consequently, transforming tundra environment into shrubs and forest vegetation. The higher amount of biomass in the shrubs and trees will increase

terrestrial carbon sink providing means for climate mitigation. Forest ecosystems can additionally enhance carbon uptake via complex atmosphere-biosphere feedback mechanisms (Kulmala et al., 2013; Kalliokoski et al., 2019). However, there can be also adverse effects. The forests and woodlands are a habitat of ticks and tree line advance could bring diseases, such as borreliosis and encephalitis, to previously non-endemic areas.

Shrubification of tundra has been reported in many recent studies (Myers-Smith and Hik, 2018; Maliniemi et al., 2018;

Bjorkman et al., 2020) and it has been associated with the greening of the Arctic (Myers-Smith et al., 2020). However, there is no general agreement regarding tree propagation. Modelling studies based on abiotic factors predict tree-line advance to the north (Tchebakova et al., 2010; Kaplan and New, 2006; Aakala et al., 2014). Nevertheless, this advance has been only partially confirmed by observations (Harsch and Bader, 2011) and observed rates are extremely low as compared to theoretical predictions (Van Bogaert et al., 2011). On the contrary, the study based on the forest inventories from the eastern United States

shows that forested areas shrink rather than expand on most plots at their range limits (Zhu et al., 2012). It was hypothesized that the biotic factors have a significant effect together with the abiotic climatic factors (Woodward et al., 2004). Availability of seeds, germination success and presence of major species competing with newly establishing plants can all influence propagation of the tree-line. A recent study suggests that the shift of biomes occurs episodically and requires a disturbance (Renwick and Rocca, 2015). In northern latitudes (boreal, subarctic and arctic areas), wildfires and grazing by reindeer are the two main

disturbances that really influence the vegetation structure and dynamics (Köster et al., 2013; Narita et al., 2015), which in turn induce shifts in ecosystem processes, e.g. nutrient cycles and ecological interactions.

The reindeer regulate the abundance of species and community composition via grazing. Both observations and field experiments in tundra show different response to warming with and without large herbivores (Post and Pedersen, 2008; Olofsson et al., 2009), manifested by enhanced growth of deciduous shrubs on the sites not affected by grazing. However, evergreen

shrubs and trees demonstrate an opposite trend towards increased growth at the sites exposed to grazing (Bernes et al., 2015). The effect of wildfires on vegetation shift in tundra is less studied. Fire has been found to reduce the cover of lichens and bryophytes (Joly et al., 2009), but enhance the growth of grasses and shrubs (Barrett et al., 2012; Narita et al., 2015). Landhausser and Wein (1993) observed that forests in the Canada's Northwest Territories advance in the forest-tundra ecotone after a strong wildfire. Long before that Sannikov et al. (1970) noted that seedlings survival is the highest on bare mineral soils, and

high intensity and severity wildfires, which remove the organic material and expose mineral soil, can favor tree survival.

Here we study the shift in vegetation based on high-resolution remote sensing data, in the territory including southern tundra, forest-tundra ecotone and northern taiga in the Nadym-Pur district of North-West Siberia. Opposite to northerly-located Yamal and Gydan peninsulas and westerly-located Priuralsky district, this territory has not yet suffered from reindeer pasture overuse (Matveev and Musaev, 2013). Therefore, grazing could be of secondary importance for the ecosystem changes in these

areas. We hypothesize, based on the study of Landhausser and Wein (1993), that forest expansion occurs mainly in the areas





affected by relatively recent wildfires, while other areas (non-disturbed background areas) demonstrate only a minor change in vegetation. Thus, we focus on the effect of wildfires on the vegetation, specifically on tundra-forest and tundra-woodlands transition. The objectives of the study are:

1) to quantify burned surface areas and assess the frequency and causes of wildfires;

2) to study the probability of dry tundra transition to woodlands and forests, separately in the fire-affected and non-disturbed background sites;

3) to study long-term time series of regional climatic factors in order to assess the possibility of shifts from certain types of biomes towards more southern ones based on abiotic factors.

## 2    Materials and methods

**2.1    Vegetation dynamics**

We used remote sensing observations and topographic maps to monitor changes in vegetation dynamics in North-West Siberia, Yamal-Nenets Autonomous district, between latitudes 65 and 67 degree (Fig. 1). The data sets used for analysis of vegetation dynamics are summarized in Table 1. We compared the data from archive imagery from the 'Corona' mission (Ruffner, 1995) and topographic maps to the state-of-the-art satellite data of super high resolution (e.g., WorldView-1,2,3,4; GeoEye-1; 'Resurs-

P' 1-3). This method is appropriate for the forest-tundra ecotone with small percentage of forest cover and spatial scales of about hundred kilometers. In forest-tundra transitional zones, it outperforms traditionally used forest masks employing optical imagery or radiolocation - Landsat (Hansen et al., 2013) and PALSAR-2/PALSAR/JERS-1 (Shimada et al., 2014). The latter, as demonstrated in Fig. 2, often result in imprecise and erroneous data (Tropek et al., 2014). The PALSAR-2 forest mask has the lowest precision. The Landsat mask has a higher precision but it is not able to reproduce the forest cover in the forest-tundra

ecotone.

Due to insufficient quality of the 'Corona' images (Frost and Epstein, 2014), we additionally used a vector topographic map (Table 1) providing information on the forest (Fig. 1) and dry tundra sites within our study areas. We studied the tundra-forest shift of the dry tundra affected by fires before 1968 and of the background tundra (not affected by fires) during the period of 1968-2018. Firstly, based on the topographic map, we determined the dry tundra areas affected by fires before 1968 (Sec.

2.2) and not affected by fires during the whole study period. Then we introduced random sample circles with a diameter 100 m within the dry tundra subsets. Then we visually compared the areas inside the circles based on 'Corona' and on modern images. We chose only those circles, for which we could confidently state that there was tundra and no trees in 1968. We focused on the vegetation shift after relatively 'old' fires (i.e., before 1968) because mature trees, unambiguously distinguished in the satellite images, had higher chances to develop during 60 years. Furthermore, we visually separated woodland (trees

cover less than 50% of the circle areas) and forest (trees cover more than 50% of the circle areas). For other fires, similar tundra-forest transition could also be expected, but visual analysis of younger trees is more challenging. Another issue is the lack of information on the vegetation state before more recent fires. Therefore, the analysis of vegetation shift after more recent fires was not performed.





To quantufy the results, we assessed the current state of vegetation using Normalized Digital Vegetation Index (NDVI).
NDVI was calculated from the visible (VIS) and near-infrared (NIR) radiation reflected by vegetation:

$$NDVI = \frac{NIR - VIS}{NIR + VIS}.\tag{1}$$

NDVI reflects the state of biomass, i.e. vegetation greening, density and development (Walker et al., 2003; Johansen and
Tømmervik, 2014; Miles and Esau, 2016). We calculated NDVI for the study area 1, separately for the sites affected by the
fires during different years and in the background.

**2.2   Wildfires**

Hot spots and areas affected by wildfires are visible in near infrared, infrared and far infrared range. We used Landsat data to
study dynamics of the burned areas with the initial state identified from 'Corona' images. The remote sensing data to find the
fire hot spots and to quantify the fire dynamics are summarized in Table 2. Mapping of the burned areas was performed by
means of object-based image analysis, successfully used for studies of landscape dynamics (Blaschke, 2010). In particular, we
applied an algorithm 'Multiresolution segmentation' in software eCognition. We studied the percentage and distribution of the
burned sites and calculated the frequency of fire return.

**2.3   Field sites**

**2.3.1   General description**

In order to study forest-tundra transition, we selected the monitoring sites satisfying two predefined criteria:
1) the sites are covered by the archive 'Corona' imagery;
2) the sites have projective forest cover less than 10% according to the Landsat forest mask (Hansen et al., 2013).

Based on these criteria, we identified three main study areas (Table 3, Fig. 1). The largest area 1 is located on the east coast
of the river Nadym, the area 2 is between the south coast of the Gulf of Ob' and river Yarudey and the area 3 is to the west
from the city of Nadym (Fig. 1). The study sites are available for detailed analyses at geoportal 'Nadym. Changes in 50 years
(1968-2018) (NC50)' (https://ageoportal.ipos-tmn.ru/nadym/). Description of the geoportal is given in Appendix A.

Northern part of the study area 1 is in a continuous permafrost zone with permafrost temperature of approximately -3°C,
while southern part and study areas 2 and 3 are in a discontinuous permafrost zone (Trofimova and Balybina, 2015). Information
on the vegetation zones of the study areas are summarized in Table 3. The vegetation zones include southern tundra, forest-
tundra ecotone and northern taiga (Ilyina et al., 1985), with the treeline crossing the study sites (Walker et al., 2005; MacDonald
et al., 2008). The forest cover in three study areas is illustrated in Fig. 1. According to the topographic map, the following
vegetation types were classified as forested areas: dense forests, shrubs in the floodplains and river deltas, forest sprouting.
Overall, 3570 km$^2$ or 17.5% of the total study area was covered by forests, with 15.5% covered by dense forest, 1.2% covered
by forest sprouting and less than 1% covered by shrubs. The dry tundra vegetation ('moss and lichen' in the topographic map)
covered 11 370 km$^2$ or 55.7% of the total study area.





The study area is an important center of oil and gas industry with the industrial city of Nadym, giant oil and gas fields Medvezhye and Yarudeyskoe as well as major gas pipelines (Yamburg-Tula, Yamburg-Yelets, SRTO-Ural, Urengoy-Center, Urengoy-Uzhgorod etc.). A new railway road 'Northern Latitudinal Railway' (Salekhard-Novy Urengoy) is currently under construction within the study area 1. Environmental impacts due to anthropogenic activities and climate change is monitored since 1970s (Matyshak et al., 2017a, b; Sizov and Lobotrosova, 2016; Kukkonen et al., 2020).

### 2.3.2    In-situ observations of permafrost state and vegetation

In August 2019, a field campaign was performed near Pangody, in the central part of the study area 1 (Fig. 1). The active layer thickness (depth of annual thaw) was measured using a metal rod: the rod was pushed vertically into the ground until it reached frozen soil. Measurements were performed at three sites: in the background area, in the area affected by fires before 1968 and in the area affected by fires in 1968 and 1988 (Table 4). In addition, the vegetation at these representative sites (Fig. 3) was

classified. The main tree species in the site were larix and spruce-larix (*Larix sibirica, Picea obovata*) in lichen woodlands with shrub-moss-lichen flat-mound mires. The sites affected by a fire before 1968 were larix and spruce-larix lichen woodlands with birch and pine. Land cover classification is reported in Table 4.

### 2.4    Calculations of climatic indices

In order to assess the climatic conditions in the region, we used meteorological data (http://meteo.ru, latest access July 2019)

from three stations, Nadym, Nyda and Novy Port (Supplementary material, SM), located along a 200 km longitudinal transection from the south (65.3N, Nadym) to the north (67.4N, Novy Port). The data set contained 52 years of quality-checked meteorological observations (from 1966 to 2018), including air temperature (time resolution 3 h), precipitation (time resolution 12 h), and cloudiness 0 to 10 (time resolution 3 h). Based on this data, we calculated mean monthly temperatures (the sum of mean daily temperatures divided by the number of days in the month), mean monthly cumulative precipitation (the sum of

daily precipitation measurements for a given month), the daily minimum and maximum temperatures for all the years.

In order to assess the climatic factors influencing the vegetation dynamics, we studied the length and mean temperature of the growing season, which we define as the period with daily mean temperatures above +5°C (Tchebakova et al., 1994). In addition, we calculated growing degree-days (GDD$_5$),

$$GDD_5 = \sum_{\overline{T} \geq 5} (\overline{T} - 5),\qquad(2)$$

where $\overline{T}$ - is the daily mean temperature [°C]. Together with the growing degree-days, another important abiotic factor controlling vegetation dynamics can be moisture availability. We calculated potential evapotranspiration (PET) from the temperature and cloudiness measurements using Bonan's modification (Bonan, 1989) of Taylor-Priestley model (Priestley and Taylor, 1972). We further used PET and precipitation time series to calculate a dryness index (DI) (Tchebakova et al., 1994), i.e. the ratio between annual PET and precipitation (P):

$$DI = \frac{PET}{P}.\qquad(3)$$





Based on GDD$_5$ and DI, we determined vegetation zones in Nyda, Nadym and Novy Port in accordance with thresholds used by Tchebakova et al. (1994) in the model SiBCliM (see Table 5).

## 3 Results

### 3.1 Temperature, precipitation and climatic indices

The mean monthly temperatures were above zero ca 4 months per year (June-September) in Novy Port and ca 4-5 months per year in Nadym (May-September) (SM, Fig. SA1). The coldest month was January with monthly mean temperature -23...-24°C, and the warmest month is July with +11...13°C.

During the last 50 years, the mean annual temperature in the study areas has increased by 3°C (Fig. 4). In addition, the daily minimum temperatures increased by 4±3°C in Novy Port (from -44 to -40°C, SM, Fig. SA2) and by 5±4°C in two other

stations (from -45 to -40°C, SM, Fig. SA3-SA4), whereas the daily maximum temperatures increased significantly only in Novy Port, by 3.5±2.2°C (from 18 to 21.5°C, SM, Fig. SA2-SA4).

The length of the growing season (SM, Fig. SA2-SA4) has increased by 24±11 days, from 72 to 97 days in Novy Port and from 97 to 121 days in Nadym, mainly due to the earlier start of the season (ca 16 days). The mean temperature has also increased, but the trend is less pronounced, $(+1...1.4) \pm 1°C$. In general, both the length and mean temperature of the growing

season exhibit great variability particularly during the recent years. Following increases in the mean temperature and the length of growing season, the growing degree-days, $GDD_5$, has increased over the last 50 years (Fig. 4).

The strongest increase in $GDD_5$ (ca 200°C days) was observed during 20 years from 1966 to 1985, after which the increase levelled off. Based on the data, the period from 1980 to 1995 seemed to be the most favorable period for a new forest establishment due to the relative stability of mean temperature and $GDD_5$. On the contrary, the last decade was characterized by

a strong variability in both mean annual temperature (peak-to-peak 3°C) and $GDD_5$ (peak-to-peak more than 400°C days), which could make seedlings' survival less probable.

Mean annual precipitation decreases from the south to the north: from 510 mm in Nadym to 330-340 mm in Novy Port (SM, Fig. SA1). Monthly variability of precipitation follows a well-pronounced seasonal cycle with a maximum in August (July-August in Nadym). Precipitation in Nadym (Fig. 5, SM, Fig. SA1) carries features of the temperate continental regime

(Chorley, 1971), whereas precipitation in Nyda and Novy Port (Fig. 5, SM, Fig. SA1) is typical for the arctic regime with the pronounced seasonal cycle with a summer maximum and annual precipitation below 400 mm. Precipitation in Nyda and Novy Port was decreasing between 1965 and 2005 (Fig. 5). Since 2005, annual precipitation in Nyda has increased and in Novy Port drastically decreased. Potential evaporation increased in all sites. Accordingly, DI increased from 0.4 to 0.5-0.6 everywhere (SM, Fig. SA5) except during a few recent years in Novy Port, when DI exceeded unity.

Based on the observed degree-days and dryness index, the vegetation class in Novy Port changed from forest-tundra to dark needled northern taiga, in Nadym – from dark needled northern taiga to dark needled middle taiga, whereas the vegetation class in Nyda remained as dark needled northern taiga (Table 5).





## 3.2 Qualitative observations of the vegetation dynamics

Based on the satellite data analysis, we identified the following typical patterns corresponding to vegetation dynamics during the last 50 years:

1. **Wildfires - combustion of the vegetation cover.** Recent wildfires (after 2016) had a negative effect on the vegetation cover. However, the vegetation after old fires is typically characterized by a higher biomass as compared to the background tundra (Sec. 3.4.2). Moreover, old fires (before 1968) in dry tundra were followed by an active afforestation. We identified several reference sites (1-8 in geoportal NC50, Appendix A) illustrating a typical succession process. Note that the reference sites mainly experienced only one fire, though a considerable part of the study area experienced fires more than once (Sec. 3.3). An example in Fig. 5 (reference site 1) shows a tundra after wildfire in 1968: the trees are not distinguishable both on the burned area and on the neighboring background area. In 2016, the burned area was covered by a forest whereas tundra vegetation remained on the background area.

2. **Background conditions – no dynamics on the low terrains with signs of frost cracking** (SM, Fig. SB1, reference sites 9, 10). Although active afforestation could be expected on the thawed cracks filled with soil, it was not observed (reference site 11 in geoportal NC50).

3. **Background conditions – active afforestation of the river valleys** (reference sites 12-16 in geoportal NC50, SM, Fig. SB2). These dynamics were observed along the lakeshores near the Gulf of Ob' (reference site 17) and on the slopes with cracks and blocks forming due to the rising air temperature and due to gravity (reference site 18 in geoportal NC50).

4. **Background conditions – afforestation of the dry sandy sites on the fluvial terraces** (reference site 19, SM, Fig. SB3). Such terraces were typically covered by pine-lichen woodlands. These sites were free of permafrost, often subject to suffosion or hollow formation under wind action. The main vegetation trends included increase in the woodland areas, partial afforestation of hollows, and active afforestation of suffosion sinkholes.

5. **Anthropogenic influence – removal of the vegetation cover**, exposed sand on the fluvial terraces. Interruption of the anthropogenic activity led to the vegetation recovery and afforestation of the affected sites. Deciduous trees such as birch and larix grew first. An example is the territory of the former railway road Salekhard – Nadym, which was out of use since 1955 (SM, Fig. SB4, reference site 20 in geoportal NC50). Some planted conifers (spruce) could be seen in SM, Fig. SB4.

## 3.3 Dynamics of fires in North-West Siberia

Weather conditions in North-West Siberia can support large-scale wildfires during summer time. The fires mainly occur in July, when an anticyclone brings warm and dry weather for longer than a week. As a result, water table lowers and the land cover becomes dry and flammable.

Burned areas according to satellite imagery from 1968-2018 are shown in Fig. 7a. We identified five major fire events. The years of these fires and satellite images used in the post-processing are reported in Table 6 (for data sources see Tables 1 and 2). The burned areas in Fig. 7a can overlap if the same site experienced more than one fire. Separate maps of the burned sites for each of five cases (Table 6) are available in geoportal NC50 (Appendix A).



The overall area of burned sites was ca 8300 km$^2$ constituting more than 40% of the total study area. Fig. 7b shows the distribution of the fraction of burned area over years based on satellite imagery. The largest fires, contributing more than 30% to the total burned area, occurred between 1989 and 2001 (mainly in 1990, Table 6). The fires in each of the periods 1953-1968, 1968-1988 and 2016-2018 contributed 10-20% of the total burned area (Fig. 7b). In addition, ca 10% of that area was

recognized both in 1988 and 2001 imagery. Overall, according to Fig. 7b, ca 80% of the burned area experienced one fire and 20% experienced fires more than once during 60 years. Approximately 17% of the fire-affected territory (1400 km$^2$) burned twice, 2.5% of the territory (200 km$^2$) burned three times and 0.2% (20 km$^2$) burned four times. These sites were characterized by a remarkably small period between consequent fires, which was estimated as 15-60 years if only major fires are accounted (Table 6).

### 3.4 Dynamics of the vegetation and its link to the fires

#### 3.4.1 Normalized Digital Vegetation Index (NDVI)

The study area 1 with territories affected by different fires and average NDVI for the burned and background areas are shown in Fig. 8. Note that here we consider the territories that experienced only one fire during the whole 60-yr period. We have not done calculations for 2016 due to a small fire-affected area (Fig. 7b). The mean NDVI of the background sites is smaller than

that of the burned sites, excluding the recently burned one. The NDVI of the recently burned site corresponded to the value reported by Ryu et al. (2018) for high intensity fires.

Given that the major fires with the impact on the imagery from 2001 occurred already in 1990, and the major fires with the impact on the imagery from 2018 occurred in 2016 (Table 6), the recovery time for the vegetation is between 3 and 29 years. Note that the fire-affected areas from 1988 and 1968 imagery had similar NDVI values suggesting that the vegetation reached

its new equilibrium state ca after 40 years, and this equilibrium state was characterized by a larger biomass than was present in the background state.

#### 3.4.2 Vegetation shift in the burned and background territories

Fig. 9 shows the results of the visual check of vegetation shift within the dry tundra affected by fires between 1953 and 1964 (Sec. 2.1, 3.3). In this time interval, 1090 km$^2$ of dry tundra was burned. The largest part of the burned tundra is in the northern

taiga zone, and relatively large part is in the transitional forest-tundra ecotone. The shift of vegetation was clearly seen in the forest-tundra areas, while it was less pronounced in the northern taiga areas. Fire impact on southern tundra zone was relatively small. We performed a similar visual analysis for the background dry tundra not affected by fires over the whole 60-yr period (Sec. 2.1). Overall, ca 6000 km$^2$ of dry tundra burned in 60-yr period, thus the total area of the background dry tundra available for analysis was ca 5300 km$^2$.

The results of vegetation dynamics analysis are summarized in Table 7 and illustrated in Fig. 10. In the absence of disturbances, only a tiny fraction (ca 5%) of southern tundra exhibited a vegetation shift from dry tundra to woodlands, whereas a somewhat larger proportion (15%) of changing vegetation was observed in the forest-tundra and northern taiga (Table 7). Note



that none of the background dry tundra sites developed a forest. On the contrary, the dry tundra sites affected by fires exhibited a significant shift towards tree-dominated vegetation. More than half of fire-effected dry tundra areas changed to forest in the

transitional forest-tundra ecotone (Table 7), and more than 10% - in northern taiga.

Finally, we assessed the link between the vegetation shift and topographic characteristics (Fig. 11). Our results show that there was a dependence on topography in the background areas: the median topography slope was close to one degree in the areas not exhibiting the vegetation shift, close to three degrees for the sites with the shift towards woodlands, and six degrees for the sites with shift towards forests. Note that there were only five points in the background areas that displayed a shift to

the forest, all of them were located in the river valleys (these points were not accounted in the statistics in Table 7). Oppositely, the slope dependence was weak in the areas affected by fires.

## 4   Discussion

### 4.1   Climatic indices and vegetation dynamics

Different climate models generally agree that the greatest warming due to the enhanced greenhouse gas emissions occurs

at northern high latitudes (Sand et al., 2016). Studies show that snowmelt and spring recovery in the Northern Hemisphere boreal and subarctic forest zone currently occurs two days/decade earlier than 40 years ago (Pulliainen et al., 2017), having a considerable effect on vegetation, but also on wildfire dynamics. Based on meteorological data, we found an elongation of the growing season by 4.5-4.8 days per decade and an increase of the growing season mean temperature by 0.2-0.3°C per decade. This means that in our study areas the long-term growing season temperature (between years 1966-2018) has increased by 1°C,

and the average length of growing season has increased by around 20 days (the growing season starts on average half a month earlier).

According to the calculated climatic indexes - growing degree-days ($GDD_5$) and dryness index (DI) (Sec. 3.1) - conditions throughout the study areas were already suitable for forests (Tchebakova et al., 1994) and the treeline should have moved further north. Interestingly, the strongest increase in $GDD_5$ was observed between years 1966 to 1985, followed by a stable

period during the 80s and 90s, and this period was the most favorable to the trees to take over the tundra areas. However, the climatic indexes do not account for underlying permafrost. Tchebakova et al. (2010) suggested that survival of tree species *Larix siberica* and *Picea obovata*, native to West Siberia, is possible for active leayer thickness (ALT) exceeding 1.5 m. Recent studies reported warming of permafrost by 0.5-1.2°C per decade in Western Siberia (Biskaborn et al., 2019). Kukkonen et al. (2020) report ALT between 1 m and 3 m in several boreholes near Nadym. Our measurements (Sec. 2.3.2), although by no

means extensive, show that the ALT increased to ca 1.5 m at tundra sites 40 years after the fire as compared to 40 cm at the background site. Due to the changes in surface energy balance, the depth of the active layer regularly increases within the first few years following fires (both in forest and tundra ecosystems), and the depth of the active layer in the burned sites may be deeper than in the unburned sites for two or three decades (Rocha and Shaver, 2011; Köster et al., 2018). Myers-Smith et al. (2019) emphasized the importance of the elongated growing season and the increased ALT in addition to summer temperature

for the vegetation shift based on the observations in the Canadian Arctic. They reported a change in a vegetation community





composition, namely an increase in shrubs and graminoids, and a decrease of bare ground, in the absence of disturbances, driven by climatic factors alone (see also Barrett et al., 2012; Narita et al., 2015).

Wildfires cause dramatic short-term changes by combusting soil organic matter and vegetation, but at the same time they create new recruitment opportunities (Bret-Harte et al., 2013). By removing the vegetation and part of the organic soil layer,
the fires create open patches on the soil, where pioneer species can start to grow. Moreover, the burned tundra is darker than the unburned area, which can accelerate tree establishment due to warmer conditions in the soil. This is supported by nutrients released by the burning biomass. Camac et al. (2017) found out that the fires increase shrub seedling survival by as much as 33-fold while warming positively affects their growth rates.

Review on remote sensing of non-disturbed Arctic permafrost areas by Jorgenson and Grosse (2016) concluded many recent
studies (2010-2015), showing the expansion of shrubs, but not the expansion of forests on Arctic areas, although the climatic conditions should be favorable. Different studies underline bigger post-fire changes in the biomass and composition of the non-vascular plant community in tundra areas (Lantz et al., 2013; Jones et al., 2013), compared to the vascular plant community (Landhausser and Wein, 1993) and in these conditions lichen biomass could take decades to centuries to recover. Landhausser and Wein (1993) studied recovery of vegetation after the strong fire in the field conditions. Within 5 years after the fire, 65%
of the area was not recovered bare ground, but 22 years after the fire the area was fully recovered. In agreement with above-mentioned study, NDVI values in our study area, corresponding to ca 29 years after the major fire of 1990 (Fig. 8), reached and slightly exceeded the background NDVI values. The smaller NDVI than the background value were reported only for areas corresponding to 3 years after the major fires of 2016. Thus, we can conclude that the post-fire vegetation recovery in our study areas took from 3 to 29 years, and the equilibrium was achieved 40 years after a fire. The increase in NDVI after fires
suggests that recovered vegetation is characterized by a larger amount of biomass, presumably due to the shift in species (Sec. 2.3.2, 3.4.1). The increased biomass and change in vegetation state (increased population of shrubs and forests) after fires is consistent with the other studies (Landhausser and Wein, 1993; Barrett et al., 2012; Narita et al., 2015).

However, much less information is available on forest expansion, and particularly on post-disturbance forest expansion in Arctic areas. Landhausser and Wein (1993) suggested that the forest advance to the north occurs gradually, initially along the
rivers where the conditions are milder. Similar to that conclusion, we found active afforestation of the background areas along the river valleys. (Landhausser and Wein, 1993) suggested that these trees provide seeds for the trees propagating northwards and the major advances of new trees happen step-wise, after the major fires. The current dynamics of the forest expansion in our study areas in the North-West Siberia seems to be aligned with this scenario. Our results based on the manual remote sensing data analysis (Sec. 3.2, 3.4) suggest that the disturbances have an important effect on the forest advance. We observed
an increase in the forest and woodland cover on the areas affected by relatively 'old' fires (60 years period). In the absence of a disturbance, the forest tends to take over tundra vegetation on the hillslopes and in the river valleys. As tundra turned into woodlands on rather moderate slopes (3 degrees), it is not likely that landslides played a definitive role. Instead, the vegetation shift may be related to the difference in insolation or hydrological regime (Walvoord et al., 2019). Notably, the topography was not an important factor for the vegetation shift in the fire-affected areas.





## 4.2 Fire sources and frequency

The distribution of burned sites between study areas 1, 2, and 3 was not homogeneous. A higher fraction within the study area 1 was affected (Fig. 7). Given the similarity of the climatic conditions, a possible explanation is difference in anthropogenic activity. Oil and gas infrastructure within the areas 2 and 3 was launched only recently, whereas geological prospecting within the area 1 started already in 1968, major construction of infrastructures occurred in 1971 and exploitation of Medvezhye field in 1977. Indeed, hot spots of the major fires were often located at the sites experiencing anthropogenic influence. As an example, hot spots of the fire of 1976 were located near the building infrastructure of Medvezhye field (SM, Fig. SC1). Some of the hot spots causing large fires of 2016 were found near temporary roads within the area 1 (SM, Fig. SC2, SC3). Another example is Yarudeyskoe oil and gas field (study area 2), discovered in 2008. There were no fires between 1968 and 2016. The infrastructure in the study area 2 was built from 2013 to 2016. Shortly after that, in July 2017, a large fire occurred nearby (SM, Fig. SC4).

However, some hot spots corresponding to the major fires of 1990 and 2016 were located far from the regions of anthropogenic influence, in the background sites. Fig. SC5 and SC6 of SM illustrate initiation and burned areas after such fires. The fire of 1990 was caused by ca 30 independent hot spots in the territory between Salekhard and Novy Urengoy, located far from the centers of anthropogenic activity. The combination of warming and increasing anthropogenic activity might shorten the intervals between the fires, as discussed below.

The mean fire return interval in boreal forests varies between 53 years and 180 years, with Siberia in the lower end and North America in the upper (de Groot et al., 2013), and in the Arctic areas it is expected to be even longer. For ca 10% of our study areas the interval between consequent fires was in the range of 15-60 years, whereas the frequency of the large-scale fire events, when more than 500 $km^2$ was burned, was once per 15-25 years. Compared to our results, the fire return interval in Siberian forests at similar latitudes but further to the east (100E), is 130 - 350 years (Kharuk et al., 2011). Note that Kharuk et al. (2011) studied a remote site, where typically lightning ignites the fire. Oppositely, our results show that the fire return period at the same latitudes can be significantly reduced, presumably due to anthropogenic influence. Concurrently, this effect could be amplified by higher temperatures and enhanced evaporation. Note that according to Fig. 5, all the years of major fires (1976, 1990, 2012 and 2016) were characterized by peaks in potential evaporation, as well as by anomalously high temperature during the growing season (SM, Fig. SA2-SA4) exceeding the mean value by 1°C.

## 5 Conclusions

Conventional forest masks that are used in satellite data analysis do not work well in the forest-tundra areas and therefore cannot be used to assess shifts from tundra to forest. We used topographic maps and high-resolution remote sensing to study vegetation dynamics in the forest-tundra zone of North-West Siberia during the last 60 years. We found that the vegetation shift from dry tundra to forest was strongly associated with fires. In the background sites experiencing no fires, only 6% of the area in southern tundra developed some trees during the 60-year period. This number increased up to 15% of the area in the forest-tundra ecotone and northern taiga. The shift in the background area was sensitive to topography, and trees appeared





mainly on the moderate slopes. In the fire-affected sites, after the same period, the tree-dominated vegetation occupied already 40-85% of the previous dry tundra in the forest-tundra ecotone and northern taiga.

Given the importance of the fires for the tundra-forest dynamics, we calculated fire frequency within the study area. The major fires, burning $600 - 2500$ km$^2$ of the study area, occurred every 15-25 years. Most of the burned area experienced fires only once during 60 years, although some parts experienced multiple burning, up to 4 times. For ca 1700 km$^2$ of the study area, the period between the consequent fires is rather short, ca 15-60 years. This was a much shorter period compared to the neighboring remote sites of Central Siberia (fire return period longer than 100 years) that are less affected by anthropogenic influence.

Monitoring of wildfires in Russia is focused on forests whereas importance of tundra fires is underestimated. This is a major overlook which might have great economic consequences taking into account that tundra fires further decrease pasture areas for reindeer, already extremely overused in North-West Siberia (by 100-150% on Yamal and Gydan peninsula according to Matveev and Musaev, 2013). In our study sites, the fires burned half of the dry tundra (6000 km$^2$) in 60 years. As concluded from our analysis, most of these areas likely shifted or will shift to woodlands and forests.

Forest and especially woodland development in previous tundra areas can have a negative impact on human health, as woodlands are the main habitat of *Ixodes* ticks, transmitters of Lyme disease (Allan et al., 2003; Brownstein et al., 2005; Sormunen et al., 2020). Therefore, in the near future, the fires via promoting woodland and forest can significantly affect incidence rates and even bring Lyme disease in previously non-endemic areas. The high population of reindeer - hosts of ticks - can be an additional factor boosting dispersion of the tick-borne diseases.

Converting the tundra to dark needle leaved forest causes an energy balance shift where especially areas with spruce have a positive (warming) feedback to climate in the future. Decrease in albedo causes increase of energy input to the surface, and partitioning of energy shifts towards sensible heat flux at forested sites (Beringer et al., 2005).

On the positive side, increase of forest areas may give a hint to enhancement of carbon sink and accordingly also of Carbon-Sink+ (Kalliokoski et al., 2019). Overall, the importance and need of detailed comprehensive long-term data is also obvious,
therefore it is important to complete and verify satellite remote sensing data by observations – to establish Station for Measuring Earth surface – Atmosphere Relations (SMEAR) (Kulmala, 2018) in North-West Siberia as a part of Pan-Eurasian Experiment (PEEX) activities (Kulmala et al., 2015; Lappalainen et al., 2016; Petäjä et al., 2020).

*Code and data availability.* Meteorological data are available on the following website: http://meteo.ru. Study areas, reference sites, burned areas and control points for vegetation change can be found in geoportal NC50 (https://ageoportal.ipos-tmn.ru/nadym/). Other data sets are
available from the authors upon request.
**Appendix A: Description of geoportal 'Nadym. Changes in 50 years (1968-2018)' (NC50)**

The geoportal 'Nadym. Changes in 50 years (1968-2018)' (NC50), available at https://ageoportal.ipos-tmn.ru/nadym/, aims to provide detailed geographical information on the study sites. There are three maps in geoportal NC50 (Fig. A1). The maps are synchronized, i.e. zoom or pan in one of the maps results in the same operation in two other maps. Each map contains a pointer
in the center.

1) Map 1 is constructed of Corona/KH-4b imagery with marked study areas and reference sites. In the upper left corner, there are zoom instruments. In the upper right corner, there is a widget containing bookmarks of the reference sites. The reference sites are marked by yellow polygons in the map 1.

2) Map 2 is constructed of super high-resolution imagery: Yandex Maps, Bing Maps or Global Forest Watch. Fire-affected
areas from different years can be overlain on this map. In addition, the field of Map 2 can be used to check the vegetation change in the control points. In the upper right corner, there is a panel of map layers.

3) Map 3 can be used as a digital elevation map (ArcticDEM) or a topographic map. In the upper right corner, there is a panel of map layers and coordinates of the pointer position (degrees-minutes-seconds). The coordinates in the decimal format are copied to the clipboard by a mouse click.

*Author contributions.* EE, OS and KK designed and conceptualized the study. OS and PT developed the geoportal, analyzed the data, prepared figures and interpreted results. NP performed on-site data acquisition. EE analysed the data, prepared figures, interpreted results and wrote the manuscript. KK interpreted results and contributed to writing discussion. AS, KK, JB, TP, SZ and MK contributed with data interpretation and writing - review and editing. All the authors commented on the manuscript.

*Competing interests.* The authors declare that they have no conflict of interest.

*Acknowledgements.* EE and TP acknowledge Academy of Finland Center of Excellence program (grant no. 307331) and Belmont Forum project Arctic Community Resilience to Boreal Environmental change: Assessing Risks from fire and disease (ACRoBEAR) via Academy of Finland, decision number 334792. OS and AS were funded by RFBR, research project 20-55-71004. PT was funded by Russian Science Foundation project 18-17-00178. MK acknowledges the Academy of Finland professor grant (no. 302958) and ATM-GTP European Research Council grant under the European Union's Horizon 2020 research and innovation program (no. 742206). OS, AS, SZ and JB acknowledge
bilateral collaboration supported by Academy of Finland grant 314 798/799 and Russian Foundation for Basic Research grants 18-55-11005 and 19-17-00209. TP acknowledges Funding through Academy of Finland ('Natural Secreted Nano Vesicles as a Source of Novel Biomass Products for Circular Economy', NANOBIOMASS, 307537). KK was supported by the Academy of Finland (Academy Research Fellow project 294600).



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





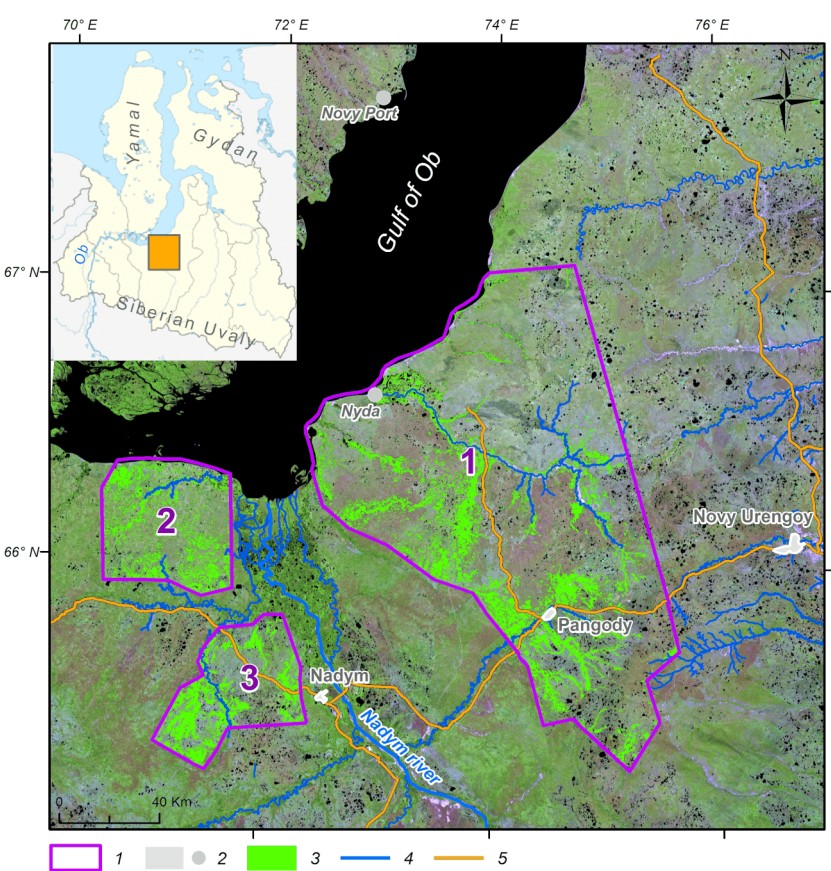

**Figure 1.** Study areas 1, 2 and 3 shown in the Landsat-7 image (2001). Forested sites within the study areas are shown according to the topographic map (Table 1). For detailed description of study sites, see Sec. 2.3.



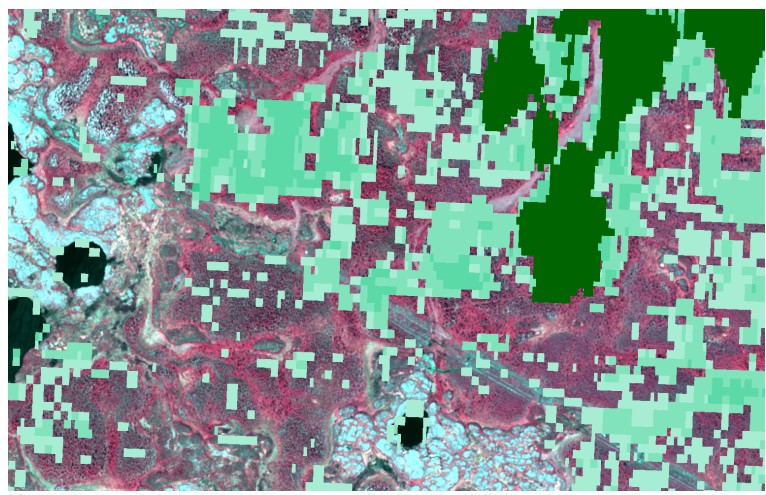

**Figure 2.** Forest masks PALSAR-2 (dark green) and Landsat-8 (light green). The forest cover is based on the image «Resurs-P» (resolution 1.65 m, synthesized in the infrared spectral range).





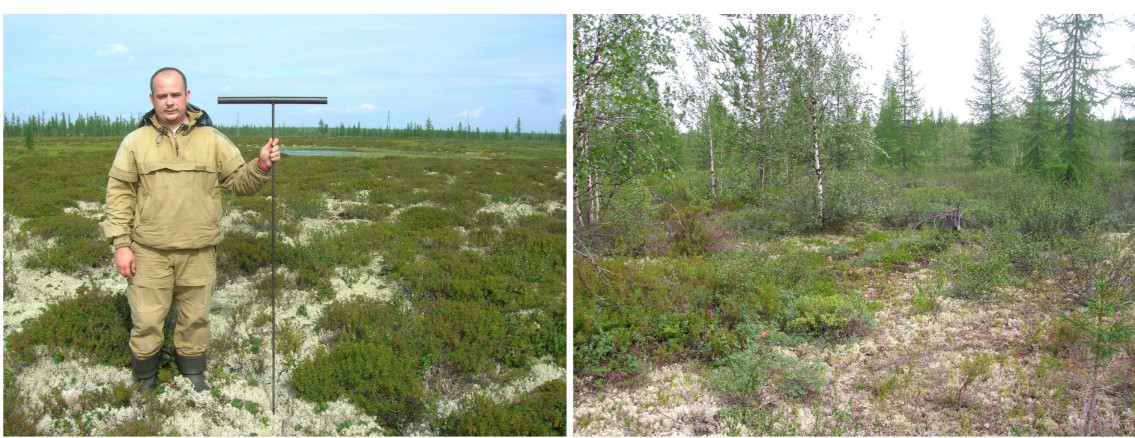

**Figure 3.** A background site (left panel) and a nearby site affected by fires before 1968 (right panel).



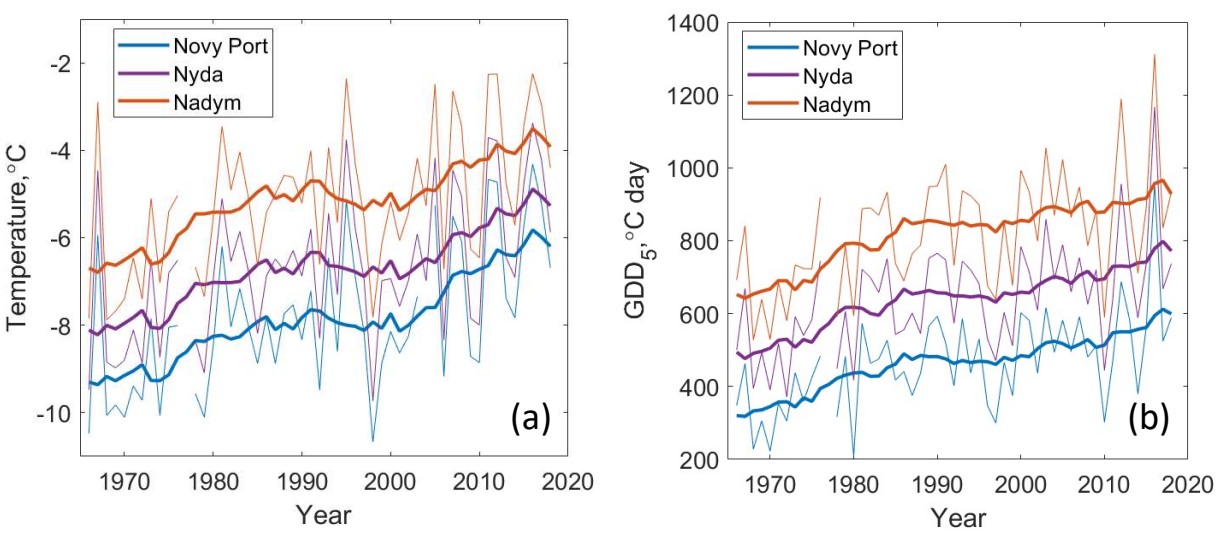

**Figure 4.** (a) Time series of mean annual temperature and (b) growing degree-days. Thick curves correspond to 10-yr running average.



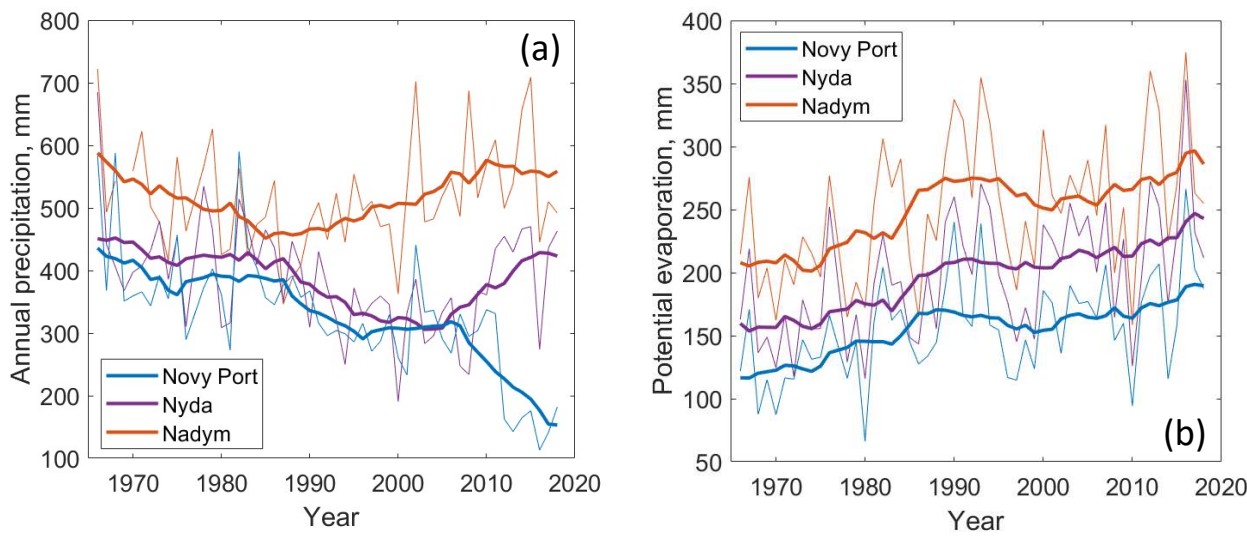

**Figure 5.** (a) Time series of annual precipitation and (b) potential evaporation. Thick curves correspond to 10-yr running average.



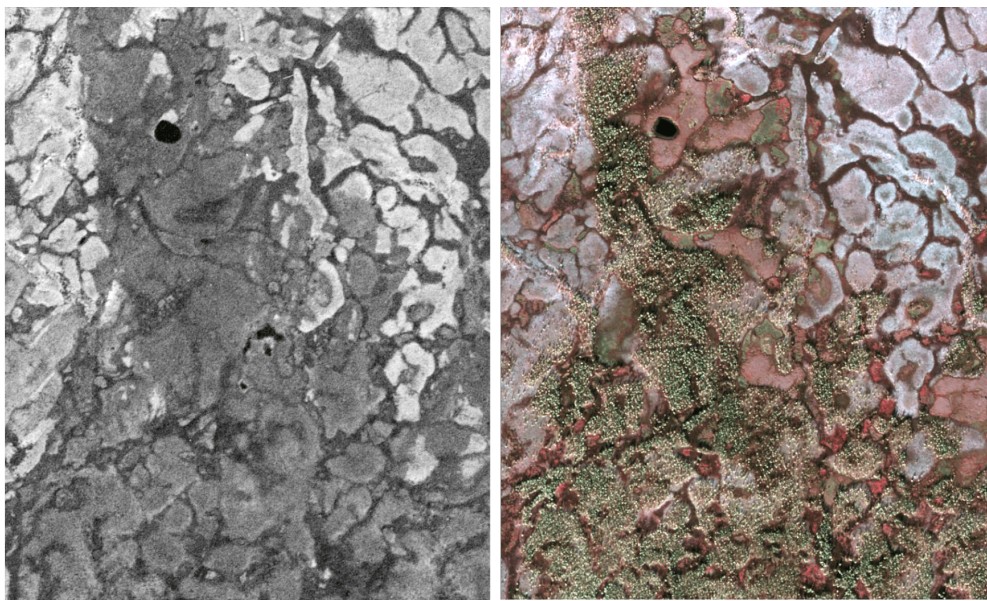

**Figure 6.** Afforestation of tundra after wildfires, reference site 1 in geoportal NC50 (left panel – Corona/KH-4b, 21/08/1968; right panel – «Resurs-P», 28/09/2016).

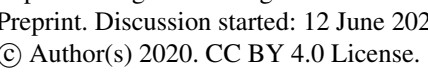



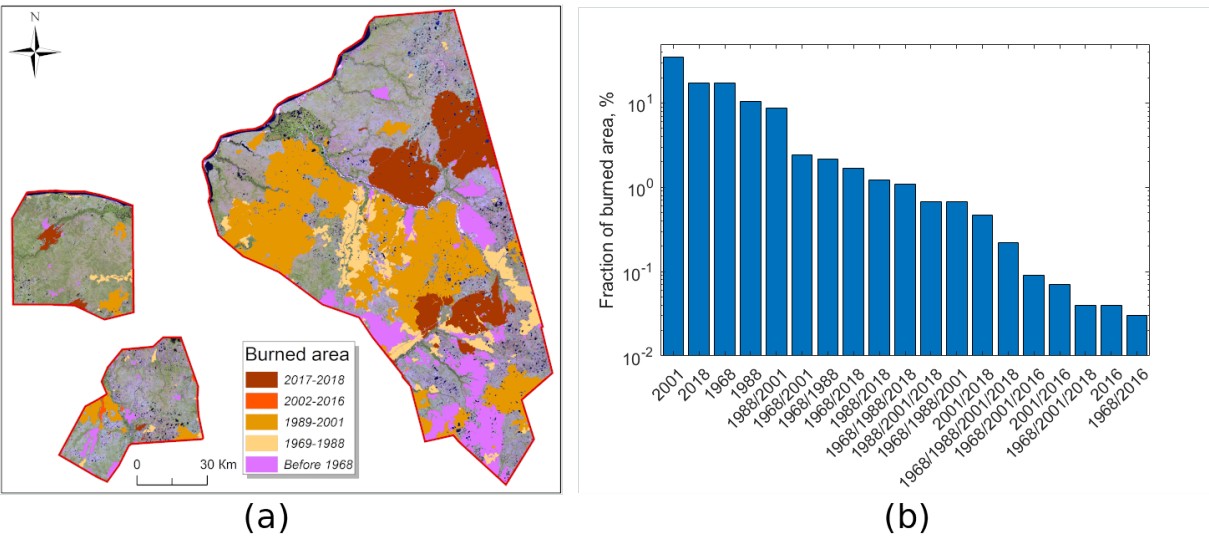

**Figure 7.** (a) Burned territories within study areas 1, 2 and 3. (b) The fraction of burned area: burned area in the satellite imagery from particular year(s) divided by the total burned area over the whole period. Multiple years in x-axis correspond to multiple fires at the same site.





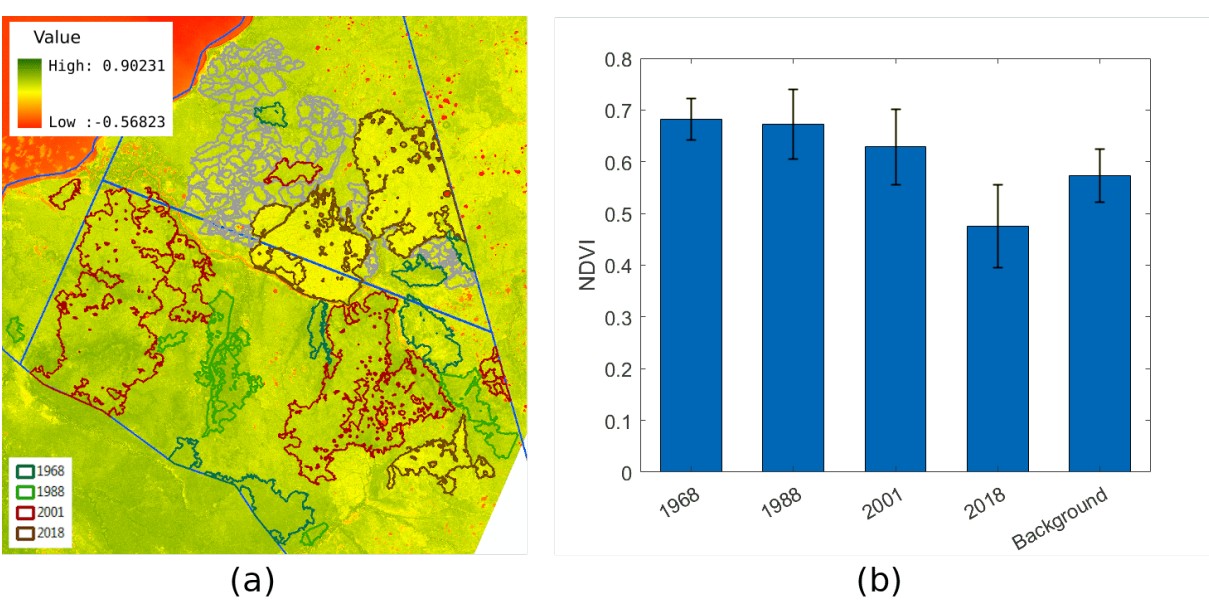

**Figure 8.** (a) The areas affected by fires used for calculations of Normalized Digital Vegetation Index (NDVI) indices. The color map shows NDVI. (b) Mean NDVI indices for the areas affected by fires in different years. Error bars show standard deviation. Note that NDVI is calculated for 2019 (Table 1).



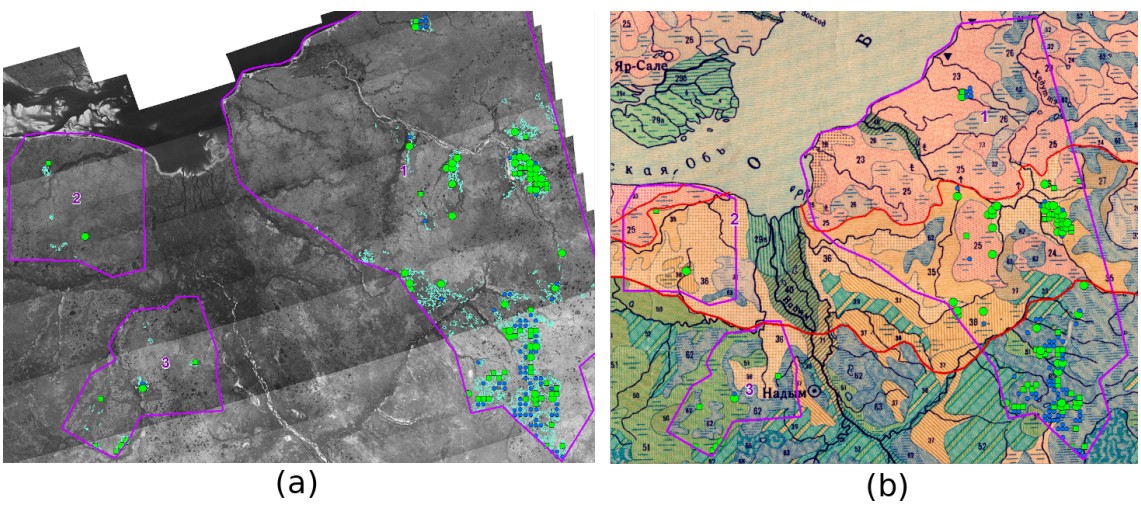

**Figure 9.** (a) Illustration of the vegetation shift experienced by dry tundra affected by fires (time interval ca 60 years). Blue circles – no shift, green squares – shift to woodlands, green circles – shift to forest. (b) Separation of southern tundra, forest-tundra ecotone and northern taiga based on the map by Ilyina et al. (1985). Boundaries are shown with red contours.





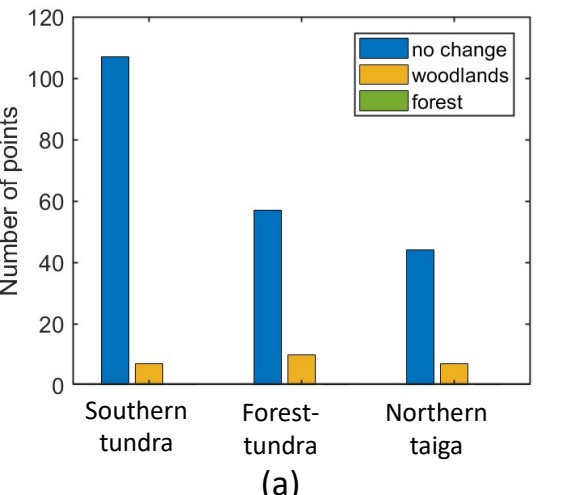 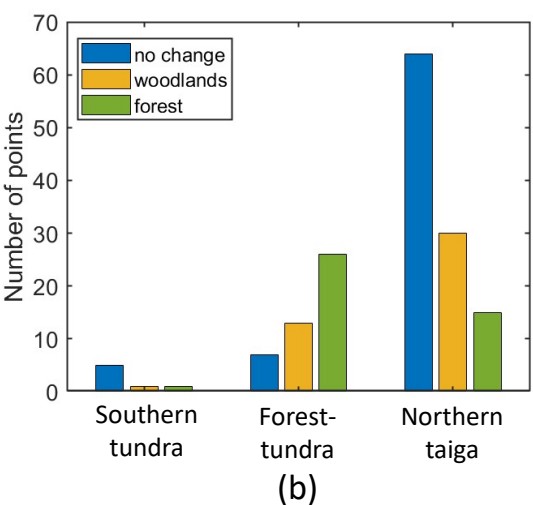

**Figure 10.** (a) Number of checkpoints for different vegetation classes, background territories. (b) Number of checkpoints for different vegetation classes, fire-affected territories. The time interval is ca 60 (55-64) years.



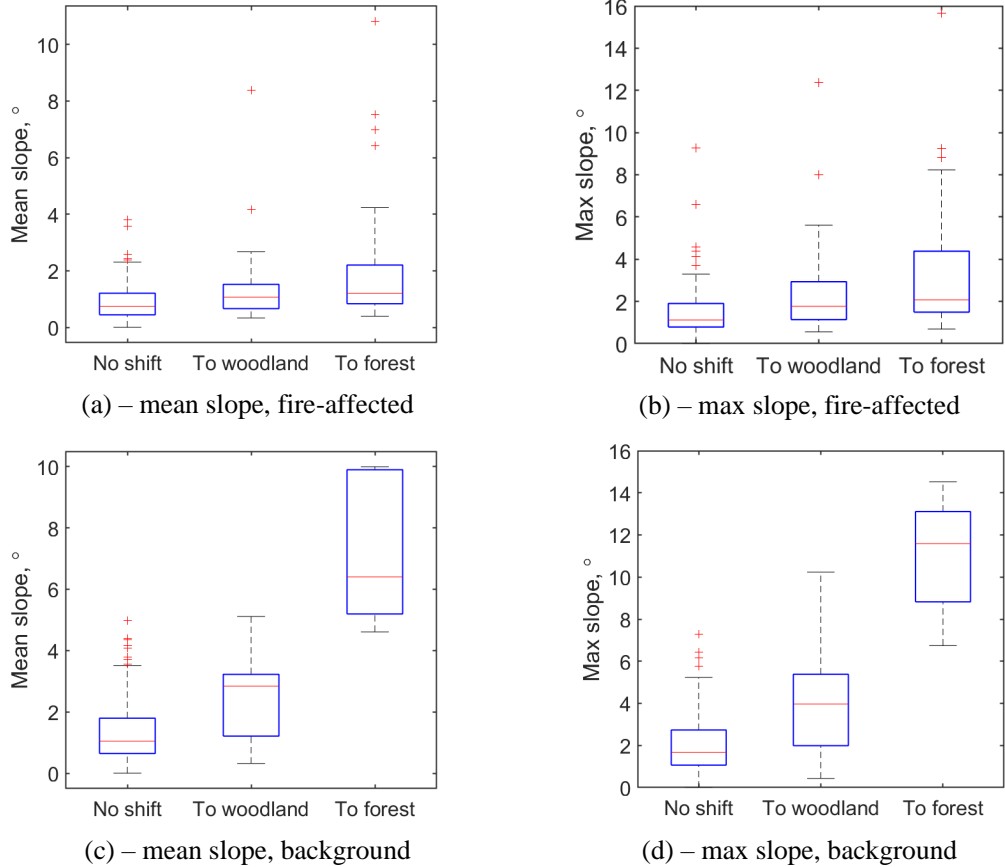

**Figure 11.** Median boxplots of mean and maximum topography slopes within the control points. The control points are circles of 50 m diameter in the centers of the check circles (Sec. 2.1). Red line - median values, bottom and top edges of boxes – 25th and 75th percentiles, whiskers 5th and 95th percentiles and red crosses - outliers.



**Table 1.** Data sets used for the analysis of vegetation dynamics

| Data set | Source | Date | Path/ map sheet number | Resolution |
|---|---|---|---|---|
| Corona/KH-4b | US Geological Survey[1] | 21.08.1968 | DS1104-2217DA(24...45) | ≈ 2 m |
| 'Resurs-P' 1,2 | Roscosmos[2] | 07.11.2016<br>04.07.2016<br>28.09.2016 | 7081_05<br>8513_02<br>9830_01 | 1.6 m |
| SPOT-6,7 | YandexMaps/ AirbusDS[3] | 2016-2017 | - | 1.5 m |
| Digital elevation model ArcticDEM, v. 3.0 | National Geographic Agency, US[4] | 2018 (survey 2010-2017) | 52_63_2_2, 52_64_1_1, 52_64_1_2, 52_64_2_2, 52_64_2_1, 52_65_1_2, 53_62_2_1, 53_62_2_2, 53_63_2_2, 53_63_1_1, 53_63_1_2, 53_63_2_1, 53_64_2_1, 53_64_1_2, 53_64_1_1, 53_65_1_1, 54_63_2_2, 54_63_2_1, 54_63_1_1, 54_64_1_2, 54_64_1_1 | 2 m |
| Topographic maps | Rosreestr[5], Ilyina et al., 1985, Supplementary material | 1968-1971 | Q-42-57,58; Q-42-59,60; Q-42-69,70; Q-42-71,72; Q-42-81,82; Q-42-83,84; Q-42-95,96; Q42-107,108; Q-43-27,28; Q-43-29,30; Q-43-37,38; Q-43-39,40; Q-43-41,42; Q-43-43,44; Q-43-49,50; Q-43-51,52; Q-43-53,54; Q-43-55,56; Q-43-61,62; Q-43-63,64; Q-43-65,66; Q-43-67,68; Q-43-73,74; Q-43-75,76; Q-43-77,78; Q-43-79,80; Q-43-85,86; Q-43-89,90; Q-43-91,92; Q-43-103,104 | 1:100 000 |
| NDVI Landsat | US Geological Survey[1] | 03.07.2019 | 16014-16013 | 30 m |

[1] https://earthexplorer.usgs.gov
[2] https://gptl.ru
[3] https://yandex.ru/maps
[4] https://arctic-nga.opendata.arcgis.com
[5] https://www.marshruty.ru/Maps/Maps.aspx





**Table 2.** Data sets used for fire analysis

| Data set | Source | Date | Path | Resolution |
|---|---|---|---|---|
| Hexagon/ KH-9 | US Geological Survey[1] | 23.07.1976 | DZB1212-500073L001001 | $\approx$ 6 m |
| Landsat-1 | | 1973 | 170014, 174013, 175013 | 60 m |
| Landsat-4,5 | | 1987-1988 | 159(013...014), 161(013...014) | 30 m |
| Landsat-7 | US Geological Survey[1] | 2000-2002 | -//- | 30 m |
| Landsat-8 | | 2016 | -//-, 160013, 160014 | 30 m |
| Landsat-8 | | 2017-2018 | -//-, 160014 | 30 m |
| Landsat-8 | | 2019 | 160013, 160014 | 30 m |
| Sentinel-2 | SentinelHub[2] and Sentinel.Playground[3] | July 2016 | 42W, 43W | 10 m |
| MODIS and VIIRS | Operational monitoring system FIRMS[4] | 2017.07.15-2017.08.05 | - | 1000 m |

[1] https://earthexplorer.usgs.gov

[2] https://scihub.copernicus.eu/dhus/#/home

[3] https://apps.sentinel-hub.com/sentinel-playground

[4] https://firms.modaps.eosdis.nasa.gov





**Table 3.** Classification of the study areas by vegetation types

| Study area | Location | Surface area (km²) | Vegetation distribution (Ilyina et al., 1985) | Vegetation distribution (Atlas of Yamal-Nenets Autonomous District, 2004) |
|---|---|---|---|---|
| 1 | triangle with vertices at (65°17'N 75°12'E), (66°30'N 72°05'E) and (67°06'N 74°40'E) | 15800 | ca 50% of dry and wet tundra (mainly northern part), 10-15% of woodlands along the rivers, 30-35% of northern taiga wetlands (mainly southern part) small areas (< 3%) along the rivers are covered by forests | ca 10% – tundra (northern part), 65% – forest-tundra, 25% – northern taiga (southern part) |
| 2 | quadrangle with vertices at (66°19'N 71°00'E), (65°55'N 71°42'E), (65°55'N 70°34'E) and (66°19'N 70°23'E) | 2500 | >50% of woodlands, ca 20% in the north – dry and wet tundra, ca 5% in the south – northern taiga | forest-tundra |
| 3 | quadrangle with vertices at (65°49'N 72°07'E), (65°26'N 72°23'E), (65°15'N 71°32'E) and (65°21'N 71°05'E) | 2000 | 20% – woodlands, 30-40% - northern taiga wetlands, 30-40% northern taiga forests. | northern taiga |

**Table 4.** Active layer thickness and vegetation based on field observations

| Site | Active layer thickness (cm) | Vegetation cover |
|---|---|---|
| 65°50'59.49"N, 74°23'03.91"E – background | 38.3 (min 38, max 40, n=3) | lichen (*Cladonia alpestris, Cladonia rangiferina, Cetraria nivalis*) – up to 40%, shrubs (*Betula nana, Ledum*) – up to 70%, cloudberry (*Rubus chamaemorus*) – up to 5% |
| 65°50'51.14"N 74°24'58.39"E – fire before 1968 and 1988 | 119.0 (min 90, max 170, n=10) | lichen (*Cladonia alpestris, Cladonia rangiferina, Cetraria nivalis*) – up to 60%, shrubs (*Betula nana, Ledum, Vaccínium uliginósum*) – up to 80%, mosses (*Polytrichum*) – up to 50% |
| 65°56'18.27"N 74°38'59.00"E – fire before 1988 | 102.0 (min 70, max 130, n=5) | lichen (*Cladonia alpestris, Cladonia rangiferina, Cetraria nivalis*) – up to 20%, shrubs (*Betula nana, Ledum, Vaccínium uliginósum*) – up to 70%, mosses (*Polytrichum*) – up to 30% |





**Table 5.** Classification of vegetation classes based on growing degree-days (GDD5) and dryness index (DI) (Tchebakova et al., 1994)

| Vegetation class | GDD5 (°C day) | DI |
|---|---|---|
| Tundra | 0-300 | < 3.3 |
| Spruce-larch forest – tundra | 300-500 | < 2.0 |
| Dark-needled northern taiga | 500-800 | < 2.3 |
| Dark-needled northern taiga | 800-1000 | < 2.3 |

**Table 6.** Relations between satellite imagery years and major fire years

| Satellite image year | 1968 | 1988 | 2001 | 2016 | 2018 |
|---|---|---|---|---|---|
| Preceding major fire year | between 1953 and 1964, source: aerial survey | 1976 | 1990 | 2012 | 2016 |

**Table 7.** Number of checkpoints (percentage of all checkpoints within a vegetation class, %) corresponding to vegetation shift in fire-affected and background dry tundra

| State of vegetation | South tundra, fires | Forest-tundra, fires | Northern taiga, fires | South tundra, background | Forest-tundra, background | Northern taiga, background |
|---|---|---|---|---|---|---|
| No shift | 5 (71.6) | 7 (15.2) | 64 (58.7) | 107 (93.9) | 57 (85.1) | 44 (86.3) |
| To woodlands | 1 (14.3) | 13 (28.3) | 30 (27.5) | 7 (6.1) | 10 (14.9) | 7 (13.7) |
| To forests | 1 (14.3) | 26 (56.5) | 15 (13.8) | 0 (0) | 0 (0) | 0 (0) |
| All | 7 (100) | 46 (100) | 109 (100) | 114 (100) | 67 (100) | 51 (100) |

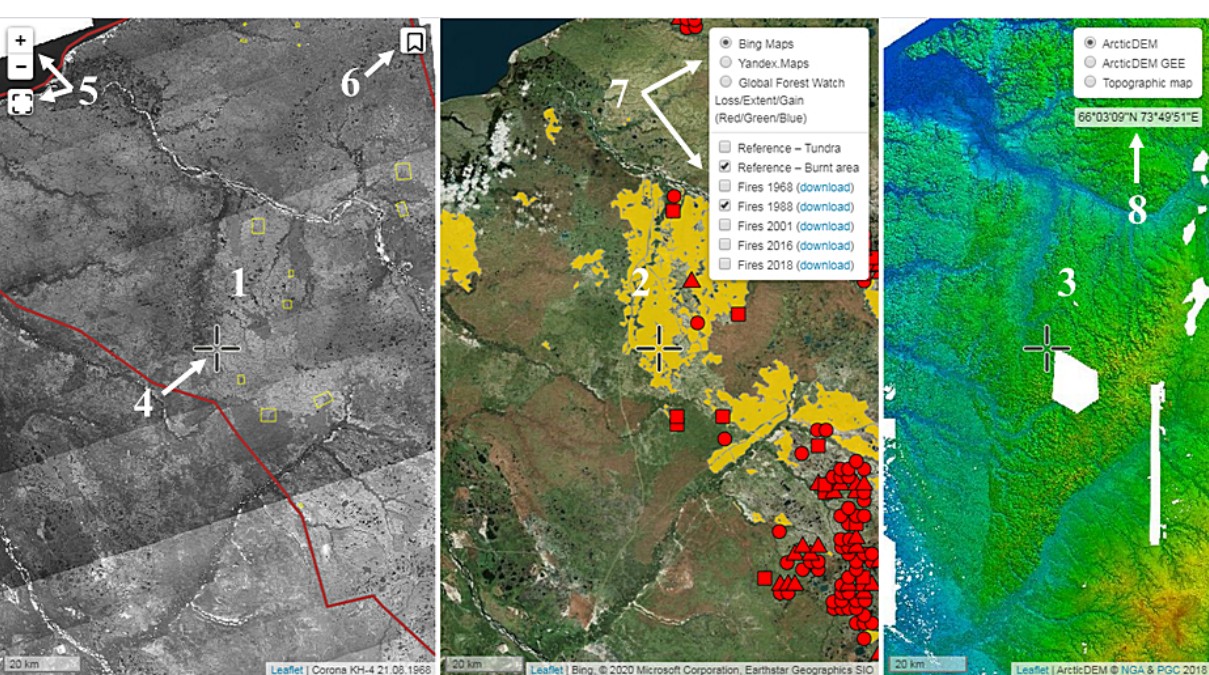

**Figure A1.** Interface of geoportal *NC* 50: 1 – map 1; 2 – map 2; 3 – map 3; 4 – map pointer; 5 – zoom; 6 – widget containing bookmarks of the reference sites; 7 – panel of base maps and layers; 8 – current coordinates.