# Peer review of "Fire and vegetation dynamics in North-West Siberia during the last 60 years based on high-resolution remote sensing"

_Biogeosciences, 2020_

## Referee Comment (RC1) · Anonymous Referee #1 · 22 Jul 2020

Dear authors,

Your manuscript addresses an important topic, highlighted again this early summer with a heat wave and related extensive fires spreading well into tundra areas across Siberia, with high estimated amounts of carbon released to the atmosphere. The topic and scientific questions formulated in the objectives are highly relevant and in the scope of BG. While the topic is important, and it is obvious that a lot of work was performed for this study, the manuscript unfortunately fails in presenting especially methods in a reproducible way. Also, the manuscript is not coherent, it looks as if different work was pieced together, but not integrated from beginning to the end.

[Figure]

A) The objectives as formulated in the introduction are not followed by a corresponding structure and sequence in the methods and results section. This makes the overall manuscript very hard to follow as a reader, as one needs to search for the corresponding information. For example, there is not a dedicated methods section that explains how the first objective (to quantify burned surface areas and assess frequency and causes of wildfires) was addressed and the sequence is changing between methods and results. Apart from structural problems, some of the objectives are not directly followed at all or in a qualitative way only. B) Methods: The manuscript contains tables with data sets, but it remains unclear which data sets were used for which objectives/results specifically, how the imagery was preprocessed given so many different data sets of highly varying spectral, spatial resolution and quality were used. Also, details on the processing of data (esp. remote sensing data, e.g. atmospheric correction) are largely missing (indicating a software without even the version or parameters used for the algorithm is not sufficient for reproducible methods). Further, there is little to no information about validation of the classification results or reference to uncertainties of results. C) Overall the presentation of the manuscript is really not sufficient – as indicated in more detail below, graphs are poor (missing legends (esp. Fig 1 & 2), scale, missing reference of Figure in main text). Consider a more rigorous selection of graphs and information displayed in tables. Also, thorough revision of language (esp. articles) and checking of consistency are needed to make this manuscript more accessible. With this list of co-authors I expected a higher quality of the overall presentation.

The following more detailed comments on suggestions for methodological improvements are based on the structure of the results section. Please note that these comments refer only to examples, a thorough overall revision of all sections is needed, esp. also to make them coherent (also applies to discussion and conclusion sections which are not covered here explicitly as changes rely on reworked methods and results):

1. Temp., precip, climatic indices - Methods for GDD5 (line 144) – provide reference for this formula. - Line 158 – what is the 3° increase based on – a trend fitted to the climate

data in Fig. 4? If yes, show the trend and related statistical information. - Data from 3 meteorological stations are presented. It remains unclear how these are linked to the 3 selected study sites as the stations are located outside of the study sites and not in obvious pairing to the 3 sites. For example on line 180 it is stated that based on the climate data analysis, the vegetation class in Novy Port changed from forest-tundra to dark needeled northern taiga – how are the climate data linked to vegetation classes, and the station data to the study sites?

2. Qualitative assessment of vegetation dynamics Overall this section is not convincing as it is largely missing a corresponding reproducible methods section. Quantitative results are hard to reach based on Corona as reference data set. But even if only qualitatively assessed, methods need to be clearly outlined. - How were these transitions qualitatively assessed? Some information can be found in section 2.1, some in the field sites general description, but nowhere is clearly formulated how the transitions were visually/qualitatively assessed, what classes were followed. Also, it remains unclear how the topographic map was used for this (does it contain forested area? Burned area?). The graphs that are mentioned to highlight how this was done are not conclusive (e.g. Fig2 misses a color legend, also it is not clear from this graph which of the layers show the most reliable forest cover, Fig SB3 – without clear indication in the imagery it is hard to understand where the active afforestation mentioned in the figure title is located – this is certainly due to the very different quality of the Corona versus Yandex map layers, but as presented does not convince the reader that this active afforestation has happened). Also, how many sites (burned and background sites) were assessed in total? Are the different conditions statistically balanced (for several of the assessed transitions only a single reference site is mentioned, does it mean that this condition was only observed once)? What was the exact sampling design? - What is active afforestation? Define in the related methods section - L. 191- fig 5 is wrongly referenced (Fig 5 displays potential evaporation, not tundra after wildfire) - L 204 – removal of vegetation cover – define in related methods section.

3. Dynamics of fires - Methods: classification to identify burned areas: how was the initial state determined in the Corona images? In table 2 you also list Sentinel, Modis and VIIRS data – how were these data used (you only mention Corona and Landsat in the fire methods section)

4. Dynamics of vegetation and fires - NDVI is NOT the normalized digital vegetation index - Which imagery was used to calculate NDVI? Any preprocessing performed? Georegistration issues discovered? Explanations on remote sensing data in methods are insufficient. - Fig 8 – what is displayed here exactly? This remains unclear based on the corresponding methods section and figure title. Is the standard deviation based on spatial variation for the background sites? How is temporal variation in NDVI of the background sites accounted for? Are the different years and the background areas statistically balanced for their size?

––––––––––––––––––––––––––––––

---

## Referee Comment (RC2) · Anonymous Referee #2 · 24 Jul 2020

The topic on itself is very interesting and highly relevant as still large uncertainties exist on the nexus climate change, fire activity, and vegetation regeneration and succession. I also appreciate the involvement of the Corona satellite, which allows for extending the observation period further back in time compared to Landsat, although I wonder how the authors come to a period of 60 years (2018-1968=50).

Nevertheless, I agree with reviewer 1 in all the points made: in its current shape the manuscript contains too many unknowns to be able to make a proper judgement of the results. I got strong feeling that many of the results (e.g. classification was based on visual interpretation). First the major issues highlighted by reviewer 1 need to be

addressed before I can provide an in-depth review of the manuscript.

---

## Author Comment (AC1) · 13 Aug 2020

**Replies to the comments of Referee 1**

We are grateful to the referee for the constructive criticism, which helped to improve the clarity of the manuscript. Please find below the replies to the specific comments and an account of the modifications implemented.

**General comments**

1. *The objectives as formulated in the introduction are not followed by a correspond-*

[Figure]

*ing structure and sequence in the methods and results section. This makes the overall manuscript very hard to follow as a reader, as one needs to search for the corresponding information.*

We apologize for this inconvenience. We have now organized the structure of the manuscript (in Result and Discussion sections) following the objectives in the introduction (Lines 59-64, p. 3). First, we calculate climatic indices to show the possibility of vegetation shift; second, we study dynamics of fires; third, we study dynamics of vegetation and its links to fires and topography.

"The main objectives of the study are: 1) to study dynamics of regional climatic factors in order to assess the possibility of vegetation shifts due to climate change, in particular tundra-forest transition; 2) to quantify burned surface areas and calculate frequency of wildfires; 3) to study the link between wildfires and dry tundra transition to woodlands and forests. Finally, we take into account physiographic characteristics of the landscape and study the effect of the topographic slope on the transition."

The structure of the manuscript is now organized as follows:

1 Introduction

2 Materials and methods

 2.1 Field sites

  2.1.1 General description

  2.1.2 In-situ observations of vegetation and permafrost state

 2.2 Calculation of climatic indices

 2.3 Wildfires

 2.4 Vegetation dynamics

 2.5 Topographic slopes

3 Results

 3.1 Temperature, precipitation and climatic indices

 3.2 Dynamics of fires

 3.3 Vegetation dynamics and its links to fires and topography

  3.3.1 Estimates of recovery time after fire using NDVI

  3.3.2 Vegetation shift using visual method, its connection to fires and topography

4 Discussion

5 Conclusion

2. *For example, there is not a dedicated methods section that explains how the first objective (to quantify burned surface areas and assess frequency and causes of wildfires) was addressed and the sequence is changing between methods and results. Apart from structural problems, some of the objectives are not directly followed at all or in a qualitative way only*

We have described how we quantified burned areas and frequency of wildfires in Methods, section 2.3 Wildfires (Lines 137-156).

In order to avoid qualitative results, we decided not to address causes of wildfires in the current manuscript and leave this topic for a more careful quantitative study in future. We have removed all the information pertaining to possible causes of fires from the manuscript. We have also removed section 'Qualitative observations of the vegetation dynamics'.

3. *Methods: The manuscript contains tables with data sets, but it remains unclear which data sets were used for which objectives/results specifically, how the imagery was preprocessed given so many different data sets of highly varying spectral, spatial resolution and quality were used. Also, details on the processing of*

*data (esp. remote sensing data, e.g. atmospheric correction) are largely miss-*
*ing (indicating a software without even the version or parameters used for the*
*algorithm is not sufficient for reproducible methods). Further, there is little to no*
*information about validation of the classification results or reference to uncertain-*
*ties of results.*

We have added the information about data sets and preprocessing of the data in
Methods (sections 2.3 Wildfires and 2.4 Vegetation dynamics).

Wildfires

Lines 126-129: 'The initial state of the study areas was obtained from 'Corona'
images. We identified 21 frames under clear-sky conditions from 21 August 1968.
Each frame consisted of 4 scanned fragments. The cropped fragments without
color correction were georeferenced to the chosen orthomosaic (SPOT layer, see
section 2.4) using polynomial method (3rd order polynomial) in software ArcGIS
(v. 10.4.1).'

Lines 132-136: 'Further, we used Landsat data to study dynamics of burned
areas. The data providing the best coverage of the study areas were available
from the following years: 1988, 2001, 2016 and 2018 (Table 4). The images were
synthesized using near- and mid-infrared channels (Landsat 5 and 7: 1.55-1.75
$\mu$m, 0.76-0.90 $\mu$m and 0.63-0.69 $\mu$m; Landsat 8: 1.57-1.65 $\mu$m, 0.85-0.88 $\mu$m,
0.64-0.67 $\mu$m) as burned areas are visible in the infrared range of wavelengths.
Landsat mosaics for all years were formed after application of color correction
using mosaic operator Blend in ArcGIS.'

Lines 137-149: 'Mapping and quantification of the burned areas were performed
by means of an object-based image analysis, successfully used for studies of
landscape dynamics (Blaschke, 2010). On the first stage, we performed seg-
mentation of mosaics using algorithm 'Multiresolution segmentation' in software
eCognition (v. 9.0). The segmentation was done using parameters 40 for Scale
and 0.5 for Color. The second stage, classification, was different for Corona and

Landsat mosaics. For Landsat mosaics, we used unsupervised classification ISODATA (15 classes, a change threshold 5%). Further, we identified visually one or two classes corresponding to burned areas. The segments containing more than 90% of pixels within these classes were identified as burned areas. In addition, we visually checked the segments with lower percentage of pixels (down to 40-50%) belonging to these classes and they were manually added to burned areas when necessary. In Corona mosaics, the spectral information was absent and we had to rely on the contrast of colors between background tundra and burned areas. In this visual check, we used two criteria. First, background tundra is lighter due to the presence of lichen in vegetation community, whereas recently burned areas are dark. Second, burned areas are characterized by well-defined boundaries often coinciding with river coastlines. An example illustrating segmentation and the visual choice of burned areas is shown in Fig 3. Calculation of the areas of segments classified as burned areas were performed using standard instrument Calculate geometry in ArcGIS.'

Vegetation dynamics

Lines 162-164: 'The initial state of vegetation was obtained from Corona imagery and topographic maps. The compilation of mosaic using Corona images is described in Section 2.3. The topographic maps were used mainly to develop the forest mask using automatic tracing in software EasyTrace, v. 8.7. The resulting vector layer was checked and corrected using Corona mosaic.'

Lines 165-169: Assessment of the current state of vegetation was based on Resurs-P and SPOT data (Table 6). Majority of the territory was covered by the mosaic of SPOT-6,7 imagery synthesized in the visible range without color correction. Ca 10% of the territory were covered by three paths of Resurs-P (the product level 2A, including four channels B, G, R and NIR). The SPOT mosaic was used as a pluggable webmap layer without additional processing. We co-registered Resurs-P data to the SPOT mosaic in ArcGIS.

Classification was performed using the visual method described in Section 2.4 'Vegetation dynamics'. Overall, visual methods are not rare in scientific studies. For example, classification of clouds performed by observer is typically taken as an etalon when automatic methods are developed. It is also not the first time when the visual methods are used for quantification of the vegetation shift. In the pioneering study, Frost and Epstein (Global Change Biology (2014) 20, 1264–1277) used a similar visual method for the same purpose. They state that 'Gambit and Corona are well suited for land-cover change studies in tundra eco-tones because tall shrubs and trees form abrupt transitions in vegetation structure that create unambiguous, readily interpreted photo-signatures. These photo-signatures result from the shadowing projected by the canopies of tall shrubs and trees, which greatly overtop tundra vegetation and create areas of high contrast in panchromatic imagery.' We have added a figure illustrating different decisions about the vegetation shift (Fig. 4) and added reference to Frost and Epstein (2014) study in the methods (Lines 160-161).

4. *Overall the presentation of the manuscript is really not sufficient – as indicated in more detail below, graphs are poor (missing legends (esp. Fig 1 2), scale, missing reference of Figure in main text). Consider a more rigorous selection of graphs and information displayed in tables. Also, thorough revision of language (esp. articles) and checking of consistency are needed to make this manuscript more accessible.*

We would like to thank reviewer for valuable comments regarding the figures and tables of the manuscript. Here is the list of modifications/adding done in the figures (numbers of figures are from the previous version of the manuscript):

Fig. 1: we added the legend. We added boundaries 'southern tundra - forest-tundra –northern taiga' in the figure.
Fig. 2: we removed the figure.
Fig. 4a: we added linear trends of temperature.

Fig. 5b: we marked years of major fires by dashed lines to emphasize connection between fires and evapotranspiration.

Fig. 6 was erroneously referenced as Fig. 5 in the previous version of the manuscript. In the present version, we have removed the section where this figure should have been referenced, as well as the figure.

Fig. 7a: increased fonts, added 'background' in the legend.

Fig. 8a: added boundaries 'southern tundra - forest-tundra –northern taiga' in the figure.

Fig. 8b is replaced by the figure with NDVI distributions.

Fig. 11: only the figures with the mean slopes are retained.

From the tables, we have removed the information related to causes of fires and classification of vegetation zones based on recent 'Atlas. . .', 2004 (Tables 1 and 4 in the current version, Tables 2 and 3 in the previous version).

We have made revision of language throughout the text and reorganized paragraphs in several sections to make them more consistent (e.g., In-situ observations of vegetation and permafrost state, Discussion). The situation with articles will be further improved if the manuscript is accepted, because all EGU journals including BG support English correction before the manuscript is published.

**Specific comments**

1. *Temp., precip, climatic indices - Methods for GDD5 (line 144) – provide reference for this formula*

   We added the reference: 'We calculated growing degree-days following Tchebakova et al. (1994)' (Line 115).

2. *Line 158 – what is the 3 °C increase based on – a trend fitted to the climate data in Fig. 4? If yes, show the trend and related statistical information.*

Yes, and we added the trends and the corresponding statistical information in Fig. 4a (Fig. 5a in the current version).

3. *Data from 3 meteorological stations are presented. It remains unclear how these are linked to the 3 selected study sites as the stations are located outside of the study sites and not in obvious pairing to the 3 sites.*

   For climate, latitudes play an important role. These three stations cover the whole range of latitudes of our study areas. The stations are located reasonably close to the study areas. One of them (Nyda) is located within study area 1, but its latitude is also close to that of the northern part of study area 2. Two stations (Novy Port and Nadym) are indeed located outside the study areas. Station Nadym is located near the southern borders of two study areas, between areas 1 and 3 (50 km from the study area 3, 150 km from the study area 1). It should be representative of the climatic conditions in the southern parts of study areas. Station Novy Port is located 70 km to the north of study area 1 and this is the closest station to the northern boundaries of the study areas. The northern part of study area 1 is located between stations Nyda and Novy Port.

   Growing degree-days, an index based on air temperature (2 m height), should be largely the same along the latitude. Precipitation can significantly vary from station to station, which is also seen in our Fig. 6, specifically on the example of Novy Port. However, the dryness index is not a limiting parameter for the vegetation shift at any of the stations and we do not expect this parameter to prevent vegetation change anywhere within our study areas.

   We added the information about location of stations in the manuscript (Lines 105-108): 'Station Nadym (65°32'N, 72°32'E, 7 m a.s.l.) is located near the southern borders of the study areas (50 km from the study area 3). Station Nyda (66°37'N, 72°57'E, 10 m a.s.l.) is located within the study area 1. Station Novy Port (67°41'N, 72°52'E, 12 m a.s.l.) is located to the north of the study areas and this is the closest station to the northern boundaries of the study areas.'

4. *For example on line 180 it is stated that based on the climate data analysis, the vegetation class in Novy Port changed from forest-tundra to dark needled northern taiga - how are the climate data linked to vegetation classes, and the station data to the study sites?*

The vegetation classes obtained from the topographic maps do not follow the classification based on the climatic indices. For example, in Nyda latitudes the maps indicate southern tundra, while the climate-based classification suggests dark needed northern taiga. This is likely the consequence of the insufficient precision of vegetation classification based on climatic indices, which can be too rough in the transitional zones. However, climatic indices illustrate that from the point of view of climate, conditions over all the study areas are suitable for forests.

5. *Qualitative assessment of vegetation dynamics. Overall this section is not convincing as it is largely missing a corresponding reproducible methods section. Quantitative results are hard to reach based on Corona as reference data set. But even if only qualitatively assessed, methods need to be clearly outlined.*

We have removed this section.

6. *How were these transitions qualitatively assessed? Some information can be found in section 2.1, some in the field sites general description, but nowhere is clearly formulated how the transitions were visually/qualitatively assessed, what classes were followed.*

We added Fig. 4 in the current manuscript to illustrate how vegetation change was assessed.

7. *Also, it remains unclear how the topographic map was used for this (does it contain forested area? Burned area?)*

It contains forested areas, and we used it to create our forest mask. We added the following information in the Section 2.4 Vegetation dynamics (Lines 163-164):

'The topographic maps were used mainly to develop the forest mask using automatic tracing in software EasyTrace, v. 8.7. The resulting vector layer was checked and corrected using Corona mosaic.'

8. *The graphs that are mentioned to highlight how this was done are not conclusive (e.g. Fig2 misses a color legend, also it is not clear from this graph which of the layers show the most reliable forest cover.*

   *Fig SB3 – without clear indication in the imagery it is hard to understand where the active afforestation mentioned in the figure title is located – this is certainly due to the very different quality of the Corona versus Yandex map layers, but as presented does not convince the reader that this active afforestation has happened). Also, how many sites (burned and background sites) were assessed in total? Are the different conditions statistically balanced (for several of the assessed transitions only a single reference site is mentioned, does it mean that this condition was only observed once)? What was the exact sampling design? - What is active afforestation? Define in the related methods section - L. 191- fig 5 is wrongly referenced (Fig 5 displays potential evaporation, not tundra after wildfire) - L 204 – removal of vegetation cover – define in related methods section.*

   Certainly, the sites were not statistically balanced as some conditions are quite specific (e.g., river valleys) but the conclusion was never made based on a single reference site. We have removed this section and this part of Supplementary material.

9. *Dynamics of fires - Methods: classification to identify burned areas: how was the initial state determined in the Corona images?*

   Added in section 2.3 Wildfires (Lines 144-149), see also answer to Q3 of General comments.

10. *In table 2 you also list Sentinel, Modis and VIIRS data – how were these data used (you only mention Corona and Landsat in the fire methods section)*

We used these data to study qualitatively hot spots of the fires, but we removed this information from the current version of the manuscript.

11. *Dynamics of vegetation and fires - NDVI is NOT the normalized digital vegetation index*

   We apologize for this error, which was unfortunately copied several times. We changed 'digital' to 'difference' throughout the text.

12. *Which imagery was used to calculate NDVI? Any preprocessing performed? Georegistration issues discovered? Explanations on remote sensing data in methods are insufficient.*

   We added this information in Section 2.4, Vegetation dynamics (Lines 188-190): 'NDVI was calculated based on two scenes from one path of Landsat-8 from 30 June 2018 (path/row 160/013 and 160/014).We used Level 2 data (CEOS) after atmospheric correction by the standard Landsat 8 OLI Atmospheric correction algorithm (Vermote et al., 2016). NDVI was calculated in ArcGIS using standard tools'.

   For ArcticDEM and satellite data sets except Corona, georegistration was performed by the data owners during orthotransformation of the satellite images. For topographic maps, georegistration was performed using coordinates of the check points in the field. The rms errors of georegistration for different types of data are reported below: Resurs-P 1,2 – 5-10 m, SPOT-6,7 – 5-8 m, ArcticDEM – 5-7 m, topographic maps – 20-25 m, Landsat – below 15 m, Corona/KH-4B – 10-15 m.

   Corona images were georegistered to the mosaic of SPOT images in ArcGIS using polynomial method (3rd order). According to the data provider (CNES, National Centre for Space Studies, France), for SPOT images, the rms of georegistration is 5-8 m and this was confirmed in different studies (e.g. Parage et al. New sensors benchmark report on SPOT 7, 2014).

For each of 84 frames of 'Corona', there is a number of check points (see Fig. 1) for which we estimated rms errors along the latitude (NS), along the longitude (EW) and 2D rms error. For all the frames, the rms error was below 10-15 m.

13. *Fig 8 – what is displayed here exactly? This remains unclear based on the corresponding methods section and figure title. Is the standard deviation based on spatial variation for the background sites? How is temporal variation in NDVI of the background sites accounted for? Are the different years and the background areas statistically balanced for their size?*

Fig. 8a shows the distribution of NDVI over the study area 1 based on the Landsat data from 30 July 2018. The curves of different colors indicate boundaries of the background tundra sites and burned tundra sites detected in the Landsat mosaics from different years (see legend). Fig. 8b showed mean NDVI indices and the standard deviations calculated for the background areas and areas burned in different years as indicated in panel (a). The standard deviation was based on the spatial variation of NDVI (determined by the number of Landsat pixels within each area) and temporal dynamics was not accounted for.

We addressed interannual temporal dynamics of NDVI using Landsat 8 data from 30 June 2018, in addition to Landsat data from 03 July 2019 we used in the previous version of the manuscript. Instead of mean values and standard deviations, in the current version of the menuscript we used distributions of NDVI (Sec. 3.3.1). In order to make the study areas more balanced, we merged the data sets for 1968 and 1988, for which we might expect that vegetation recovered after fires, based on the mean NDVI values. Currently, all study areas are larger than 1000 km$^2$ (1968+1988 – 1265 (807+458) km$^2$, 2001 – 2320 km$^2$, 2018 – 1331 km$^2$, background – 1109 km$^2$). The figures below (Fig. 8 in the new version of the manuscript, Fig. 2 in this document) show the distributions of NDVI for the burned areas detected in 1968+1988, 2001, 2018, and for background areas. Using the dates of major fires, we can assume that 1968+1988 data show

the state of vegetation in the site burned more than 42 years ago, 2001 data – 28 years ago, 2018 data – 2 years ago and background data refer to tundra not affected by fires during the whole study period.

The distributions are close to Gaussian ones for the background tundra and recently burned sites. Interestingly, the distributions are bimodal in the sites burned 28 and >42 years ago and they moved to higher NDVI values. We have fitted the distributions by the sums of two Gaussian functions and determined mean values and standard deviations for all the peaks. We found that bimodal distributions have almost the same two peaks (Table 1 in this document, Table 7 in the new version of manuscript). However, in the distribution from the sites burned 28 years ago, the peak with lower NDVI is more pronounced as compared to the peak with higher NDVI. Oppositely, in the sites burned >42 years ago, the peak with higher NDVI becomes more pronounced.

Furthermore, we used the mean values and standard deviations to identify vegetation associated with the peaks of the distributions in the satellite images. For illustration, we chose an image containing all representative examples of vegetation (Fig. 3 in this document). Green color in Fig 3, right panel, indicates the pixels which have NDVI in the interval (NDVI$_{max,2}$-$\sigma_2$; NDVI$_{max,2}$+$\sigma_2$) corresponding to the upper peak of the distribution based on the data from 1968+1988. This peak is mainly associated with forests. The lower peak (pixels in blue color, Fig. 3b), as can be seen from Fig. 3, corresponds to woodlands and tundra. This lower peak has a large intersection with the peaks in the distributions based on the data after recent fire and background data. However, interestingly, there is a significant decrease in the pixels with NDVI below ca 0.52 in the sites burned more than 28 years ago. These sites are marked in pink in Fig. 3, right panel. They correspond to the tundra sites lightest in color due to the presence of lichen in the vegetation community. The fraction of such pixels decreases in bimodal distributions, meaning that lichen does not recover to previous state. Instead,

bimodal distributions gain a large fraction of pixels with high NDVI corresponding to forests, absent in background tundra.

The distributions of NDVI are similar both for 2018 and 2019, confirming our results, but in 2019, the upper peak in the bimodal distributions is less pronounced (Fig. 4 in the current document). This could be due to the different phenological states of vegetation, dependent on temperature and precipitation from year to year. The peaks corresponding to tundra vegetation and woodlands almost did not change their position in both figures, but the peaks of the distribution after the recent fire (2018 in the legends) and the forest peaks are higher in 2018.

Finally, we estimated the fraction of forest in bimodal distributions. We used NDVI data from 2018, as the separate peaks are better pronounced in bimodal distributions. Using standard deviations of the two peaks, the boundary approximately separating forest peak from tundra peak corresponds to the threshold value of NDVI=0.73. We assume that pixels with NDVI $> 0.73$ represent mainly forest, and pixels with NDVI$<0.73$ – tundra and woodlands. We integrated the distribution to find the fraction of pixels with NDVI$>0.73$ in the total number of pixels. In the areas burned 28 years ago, forests occupy 24% of the total area. In the areas burned more than 42 years ago, the forest fraction increases to 55% of the total area. This number is comparable to our estimates of the vegetation shift within forest-tundra zone (56%) and exceeds the estimates for the northern taiga zone (14%).

While precise estimates of forest fraction based on NDVI can be challenging, the main results following from the distributions in Fig. 2b can be summarized as follows:

1. The NDVI distributions based on the data from background tundra and areas burned 2 years ago are predominantly unimodal, whereas the distributions based on the data from areas burned 28 years ago and earlier are bimodal.

2. The low NDVI pixels corresponding to vegetation communities in tundra characterized by relatively high amounts of lichen and thus having white color in the images largely disappear from the distributions corresponding to vegetation communities recovered after fires.

3. Instead, the new state of vegetation recovered after fires is characterized by a higher mean NDVI due to a new peak associated with forest. The fraction of pixels representing forest increases with time after the fire.

We thank again the referee for the useful suggestions. We hope that the present manuscript addresses all the comments raised.

**Table 1.** Best fit parameters of the NDVI distributions in Fig. 2b by the sum of two Gaussian functions (see manuscript for the formula).

| Year | $A_1$ | $NDVI_{max,1}$ | $\sigma_1$ | $A_2$ | $NDVI_{max,2}$ | $\sigma_2$ |
|---|---|---|---|---|---|---|
| 1968+1988 | 489 | 0.66 | 0.07 | 496 | 0.78 | 0.04 |
| 2001 | 1459 | 0.63 | 0.05 | 598 | 0.75 | 0.06 |
| 2018 | 367 | 0.54 | 0.09 | 554 | 0.55 | 0.05 |
| Background | 420 | 0.60 | 0.04 | 468 | 0.58 | 0.06 |

---

## Author Comment (AC3) · 14 Aug 2020

**Replies to the comments of Referee 1**

We are grateful to the referee for the constructive criticism, which helped to improve the clarity of the manuscript. Please find below the replies to the comments and an account of the modifications implemented.

**General comments**

1. *The objectives as formulated in the introduction are not followed by a corresponding structure and sequence in the methods and results section. This makes the overall manuscript very hard to follow as a reader, as one needs to search for the corresponding information.*

We apologize for this inconvenience. We have now organized the structure of the manuscript (in Result and Discussion sections) following the objectives in the introduction (Lines 59-64, p. 3). First, we calculate climatic indices to show the possibility of vegetation shift; second, we study dynamics of fires; third, we study dynamics of vegetation and its links to fires and topography. "The main objectives of the study are:
1) to study dynamics of regional climatic factors in order to assess the possibility of vegetation shifts due to climate change, in particular tundra-forest transition;
2) to quantify burned surface areas and calculate frequency of wildfires;
3) to study the link between wildfires and dry tundra transition to woodlands and forests.
Finally, we take into account physiographic characteristics of the landscape and study the effect of the topographic slope on the transition."

The structure of the manuscript is now organized as follows:
1 Introduction
2 Materials and methods
 2.1 Field sites
   2.1.1 General description
   2.1.2 In-situ observations of vegetation and permafrost state
 2.2 Calculation of climatic indices
 2.3 Wildfires
 2.4 Vegetation dynamics
 2.5 Topographic slopes
3 Results
 3.1 Temperature, precipitation and climatic indices
 3.2 Dynamics of fires
 3.3 Vegetation dynamics and its links to fires and topography
   3.3.1 Estimates of recovery time after fire using NDVI
   3.3.2 Vegetation shift using visual method, its connection to fires and topography
4 Discussion
5 Conclusion

2. *For example, there is not a dedicated methods section that explains how the first objective (to quantify burned surface areas and assess frequency and causes of wildfires) was addressed and the sequence is changing between methods and results. Apart from structural problems, some of the objectives are not directly followed at all or in a qualitative way only*

We have described how we quantified burned areas (please find in the answer to the next question) and frequency of wildfires in Methods, section 2.3 Wildfires (Lines 137-156).

'We studied the percentage and distribution of the burned sites and calculated the frequency of fire return. Corona and Landsat images showed that some years were characterized by particularly large-scale fires in the study areas (see an example for 1990 in SM, Fig. SB1). These years are referred to as the years of major fires. The burned areas can be detected in the satellite images during a few years after the fire. Landsat mosaics from 1988, 2001, 2016 and 2018 largely reflect the state of the study areas after the major fire years 1976, 1990, 2012 and 2016 (Table 5). Burned areas in Corona mosaic from 1968 were partially dated back to the fires in the period 1953-1964 based on geological surveys and early Corona images (the sources are listed below Table 5). The period between fires was calculated as the difference in years between the major fires.'

In order to avoid qualitative results, we decided not to address causes of wildfires in the current manuscript and leave this topic for a more careful quantitative study in future. We have removed all the information pertaining to possible causes of fires from the manuscript. We have also removed section 'Qualitative observations of the vegetation dynamics'.

3. *Methods: The manuscript contains tables with data sets, but it remains unclear which data sets were used for which objectives/results specifically, how the imagery was preprocessed given so many different data sets of highly varying spectral, spatial resolution and quality were used. Also, details on the processing of data (esp. remote sensing data, e.g. atmospheric correction) are largely missing (indicating a software without even the version or parameters used for the algorithm is not sufficient for reproducible methods). Further, there is little to no information about validation of the classification results or reference to uncertainties of results.*

We have added the information about data sets and preprocessing of the data in Methods (sections 2.3 Wildfires and 2.4 Vegetation dynamics).

Wildfires
*Lines 126-129*: 'The initial state of the study areas was obtained from 'Corona' images. We identified 21 frames under clear-sky conditions from 21 August 1968. Each frame consisted of 4 scanned fragments. The cropped fragments without color correction were georeferenced to the chosen orthomosaic (SPOT layer, see section 2.4) using polynomial method (3rd order polynomial) in software ArcGIS (v. 10.4.1).'
*Lines 132-136*: 'Further, we used Landsat data to study dynamics of burned areas. The data providing the best coverage of the study areas were available from the following years: 1988, 2001, 2016 and 2018 (Table 4). The images were synthesized using near- and mid-infrared channels (Landsat 5 and 7: 1.55-1.75μm, 0.76-0.90μm and 0.63-0.69μm; Landsat 8: 1.57-1.65μm, 0.85-0.88μm, 0.64-0.67μm) as burned areas are visible in the infrared range of wavelengths. Landsat mosaics for all years were formed after application of color correction using mosaic operator Blend in ArcGIS.'
*Lines 137-149*: 'Mapping and quantification of the burned areas were performed by means of an object-based image analysis, successfully used for studies of landscape dynamics (Blaschke, 2010). On the first stage, we performed segmentation of mosaics using algorithm 'Multiresolution segmentation' in software eCognition (v. 9.0). The segmentation was done using parameters 40 for Scale and 0.5 for Color. The second stage, classification, was different for Corona and Landsat mosaics. For Landsat mosaics, we used unsupervised classification ISODATA (15 classes, a change threshold 5%). Further, we identified visually one or two classes corresponding to burned areas.

The segments containing more than 90% of pixels within these classes were identified as burned areas. In addition, we visually checked the segments with lower percentage of pixels (down to 40-50%) belonging to these classes and they were manually added to burned areas when necessary. In Corona mosaics, the spectral information was absent and we had to rely on the contrast of colors between background tundra and burned areas. In this visual check, we used two criteria. First, background tundra is lighter due to the presence of lichen in vegetation community, whereas recently burned areas are dark. Second, burned areas are characterized by well-defined boundaries often coinciding with river coastlines. An example illustrating segmentation and the visual choice of burned areas is shown in Fig 3. Calculation of the areas of segments classified as burned areas were performed using standard instrument Calculate geometry in ArcGIS.'

Vegetation dynamics
*Lines 162-164:* 'The initial state of vegetation was obtained from Corona imagery and topographic maps. The compilation of mosaic using Corona images is described in Section 2.3. The topographic maps were used mainly to develop the forest mask using automatic tracing in software EasyTrace, v. 8.7. The resulting vector layer was checked and corrected using Corona mosaic.'
*Lines 165-169:* Assessment of the current state of vegetation was based on Resurs-P and SPOT data (Table 6). Majority of the territory was covered by the mosaic of SPOT-6,7 imagery synthesized in the visible range without color correction. Ca 10% of the territory were covered by three paths of Resurs-P (the product level 2A, including four channels B, G, R and NIR). The SPOT mosaic was used as a pluggable webmap layer without additional processing. We co-registered Resurs-P data to the SPOT mosaic in ArcGIS.

Classification was performed using the visual method described in Section 2.4 'Vegetation dynamics'. Overall, visual methods are not rare in scientific studies. For example, classification of clouds performed by observer is typically taken as an etalon when automatic methods are developed. It is also not the first time when the visual methods are used for quantification of the vegetation shift. In the pioneering study, Frost and Epstein (Global Change Biology (2014) 20, 1264–1277) used a similar visual method for the same purpose. They state that 'Gambit and Corona are well suited for land-cover change studies in tundra ecotones because tall shrubs and trees form abrupt transitions in vegetation structure that create unambiguous, readily interpreted photo-signatures. These photo-signatures result from the shadowing projected by the canopies of tall shrubs and trees, which greatly overtop tundra vegetation and create areas of high contrast in panchromatic imagery.'
We have added a figure illustrating different decisions about the vegetation shift (Fig. 4) and added reference to Frost and Epstein (2014) study in the methods (Lines 160-161).

4. *Overall the presentation of the manuscript is really not sufficient – as indicated in more detail below, graphs are poor (missing legends (esp. Fig 1 & 2), scale, missing reference of Figure in main text). Consider a more rigorous selection of graphs and information displayed in tables. Also, thorough revision of language (esp. articles) and checking of consistency are needed to make this manuscript more accessible.*

We would like to thank reviewer for valuable comments regarding the figures and tables of the manuscript. Here is the list of modifications/adding done in the figures (numbers of figures are from the previous version of the manuscript):
Fig. 1: we added the legend. We added boundaries 'southern tundra - forest-tundra –northern taiga' in the figure.
Fig. 2: we removed the figure.
Fig. 4a: we added linear trends of temperature.

Fig. 5b: we marked years of major fires by dashed lines to emphasize connection between fires and evapotranspiration.

Fig. 6 was erroneously referenced as Fig. 5 in the previous version of the manuscript. In the present version, we have removed the section where this figure should have been referenced, as well as the figure.

Fig. 7a: increased fonts, added 'background' in the legend.

Fig. 8a: added boundaries 'southern tundra - forest-tundra –northern taiga' in the figure.

Fig. 8b is replaced by the figure with NDVI distributions.

Fig. 11: only the figures with the mean slopes are retained.

From the tables, we have removed the information related to causes of fires and classification of vegetation zones based on recent 'Atlas…', 2004 (Tables 1 and 4 in the current version, Tables 2 and 3 in the previous version).

We have made revision of language throughout the text and reorganized paragraphs in several sections to make them more consistent (e.g., In-situ observations of vegetation and permafrost state, Discussion). The situation with articles will be further improved if the manuscript is accepted, because all EGU journals including BG support English correction before the manuscript is published.

**Specific comments**

1. *Temp., precip, climatic indices - Methods for GDD5 (line 144) – provide reference for this formula.*

We added the reference: 'We calculated growing degree-days following Tchebakova et al. (1994)' (Line 115).

> *Line 158 – what is the 3∘ increase based on – a trend fitted to the climate data in Fig. 4? If yes, show the trend and related statistical information.*

Yes, and we added the trends and the corresponding statistical information in Fig. 4a (Fig. 5a in the current version).

> *- Data from 3 meteorological stations are presented. It remains unclear how these are linked to the 3 selected study sites as the stations are located outside of the study sites and not in obvious pairing to the 3 sites.*

For climate, latitudes play an important role. These three stations cover the whole range of latitudes of our study areas. The stations are located reasonably close to the study areas. One of them (Nyda) is located within study area 1, but its latitude is also close to that of the northern part of study area 2. Two stations (Novy Port and Nadym) are indeed located outside the study areas. Station Nadym is located near the southern borders of two study areas, between areas 1 and 3 (50 km from the study area 3, 150 km from the study area 1). It should be representative of the climatic conditions in the southern parts of study areas. Station Novy Port is located 70 km to the north of study area 1 and this is the closest station to the northern boundaries of the study areas. The northern part of study area 1 is located between stations Nyda and Novy Port.

Growing degree-days, an index based on air temperature (2 m height), should be largely the same along the latitude. Precipitation can significantly vary from station to station, which is also seen in

our Fig. 6, specifically on the example of Novy Port. However, the dryness index is not a limiting parameter for the vegetation shift at any of the stations and we do not expect this parameter to prevent vegetation change anywhere within our study areas.

We added the information about location of stations in the manuscript (Lines 105-108): 'Station Nadym (65°32'N, 72°32'E, 7 m a.s.l.) is located near the southern borders of the study areas (50 km from the study area 3). Station Nyda (66°37'N, 72°57'E, 10 m a.s.l.) is located within the study area 1. Station Novy Port (67°41'N, 72°52'E, 12 m a.s.l.) is located to the north of the study areas and this is the closest station to the northern boundaries of the study areas.'

> *For example on line 180 it is stated that based on the climate data analysis, the vegetation class in Novy Port changed from forest-tundra to dark needled northern taiga - how are the climate data linked to vegetation classes, and the station data to the study sites?*

The vegetation classes obtained from the topographic maps do not follow the classification based on the climatic indices. For example, in Nyda latitudes the maps indicate southern tundra, while the climate-based classification suggests dark needed northern taiga. This is likely the consequence of the insufficient precision of vegetation classification based on climatic indices, which can be too rough in the transitional zones. However, climatic indices illustrate that from the point of view of climate, conditions over all the study areas are suitable for forests.

> 2. *Qualitative assessment of vegetation dynamics. Overall this section is not convincing as it is largely missing a corresponding reproducible methods section. Quantitative results are hard to reach based on Corona as reference data set. But even if only qualitatively assessed, methods need to be clearly outlined.*

We have removed this section.

> *- How were these transitions qualitatively assessed? Some information can be found in section 2.1, some in the field sites general description, but nowhere is clearly formulated how the transitions were visually/qualitatively assessed, what classes were followed.*

We added Fig. 4 in the current manuscript to illustrate how vegetation change was assessed.

> *Also, it remains unclear how the topographic map was used for this (does it contain forested area? Burned area?).*

It contains forested areas, and we used it to create our forest mask. We added the following information in the Section 2.4 Vegetation dynamics (Lines 163-164):
'The topographic maps were used mainly to develop the forest mask using automatic tracing in software EasyTrace, v. 8.7. The resulting vector layer was checked and corrected using Corona mosaic.'

> *The graphs that are mentioned to highlight how this was done are not conclusive (e.g. Fig2 misses a color legend, also it is not clear from this graph which of the layers show the most reliable forest cover.*

> *Fig SB3 – without clear indication in the imagery it is hard to understand where the active afforestation mentioned in the figure title is located – this is certainly due to the very different quality of the Corona versus Yandex map layers, but as presented does not*

*convince the reader that this active afforestation has happened). Also, how many sites (burned and background sites) were assessed in total? Are the different conditions statistically balanced (for several of the assessed transitions only a single reference site is mentioned, does it mean that this condition was only observed once)? What was the exact sampling design? - What is active afforestation? Define in the related methods section - L. 191- fig 5 is wrongly referenced (Fig 5 displays potential evaporation, not tundra after wildfire) - L 204 – removal of vegetation cover – define in related methods section.*

Certainly, the sites were not statistically balanced as some conditions are quite specific (e.g., river valleys) but the conclusion was never made based on a single reference site. We have removed this section and this part of Supplementary material.

3. *Dynamics of fires - Methods: classification to identify burned areas: how was the initial state determined in the Corona images?*

Added in section 2.3 Wildfires (Lines 144-149), see also answer to Q3 of General comments.

*In table 2 you also list Sentinel, Modis and VIIRS data – how were these data used (you only mention Corona and Landsat in the fire methods section)*

We used these data to study qualitatively hot spots of the fires, but we removed this information from the current version of the manuscript.

4. *Dynamics of vegetation and fires - NDVI is NOT the normalized digital vegetation index*

We apologize for this error, which was unfortunately copied several times. We changed 'digital' to 'difference' throughout the text.

*- Which imagery was used to calculate NDVI? Any preprocessing performed? Georegistration issues discovered? Explanations on remote sensing data in methods are insufficient. –*

We added this information in Section 2.4, Vegetation dynamics (Lines 188-190):
'NDVI was calculated based on two scenes from one path of Landsat-8 from 30 June 2018 (path/row 160/013 and 160/014).We used Level 2 data (CEOS) after atmospheric correction by the standard Landsat 8 OLI Atmospheric correction algorithm (Vermote et al., 2016). NDVI was calculated in ArcGIS using standard tools'.

For ArcticDEM and satellite data sets except Corona, georegistration was performed by the data owners during orthotransformation of the satellite images. For topographic maps, georegistration was performed using coordinates of the check points in the field. The rms errors of georegistration for different types of data are reported below:
Resurs-P 1,2 – 5-10 m,
SPOT-6,7 – 5-8 m,
ArcticDEM – 5-7 m,
topographic maps – 20-25 m,
Landsat – below 15 m,
Corona/KH-4B – 10-15 m.

Corona images were georegistered to the mosaic of SPOT images in ArcGIS using polynomial method (3rd order). According to the data provider (CNES, National Centre for Space Studies,

France), for SPOT images, the rms of georegistration is 5-8 m and this was confirmed in different studies (e.g. Parage et al. New sensors benchmark report on SPOT 7, 2014).
For each of 88 frames of 'Corona', there is a number of check points (see figure below) for which we estimated rms errors along the latitude (NS), along the longitude (EW) and 2D rms error. For all the frames, the rms error was below 10-15 m.

[Figure]

Fig. 1. Georegistration of Corona frames to the SPOT mosaic

*Fig 8 – what is displayed here exactly? This remains unclear based on the corresponding methods section and figure title. Is the standard deviation based on spatial variation for the background sites? How is temporal variation in NDVI of the background sites accounted for? Are the different years and the background areas statistically balanced for their size?*

In the previous version of the manuscript, Fig. 8a showed the distribution of NDVI over study area 1 based on the Landsat data from 3 July 2019. The curves of different colors indicated boundaries of the background tundra sites and burned tundra sites detected in the Landsat mosaics from different years (legend). Fig. 8b showed mean NDVI indices and standard deviations calculated for the background areas and the areas burned in different years as indicated in panel (a). The standard deviation was based on the spatial variation of NDVI (determined by the number of pixels with different NDVI within each area) and temporal dynamics was not accounted for. Instead of mean values and standard deviations, in the current version of the manuscript we use distributions of NDVI. We also show NDVI based on the Landsat data from 30 June 2018 instead of data from 3 July 2019 (the reason is described below). Therefore, Fig. 8 has changed (see Fig. 2 in this document).

In order to make the size of areas more balanced, we merged the data sets for 1968 and 1988, for which we might expect that vegetation recovered after fires, based on the mean NDVI values. Currently, all study areas are larger than 1000 km$^2$ (1968+1988 – 1265 (807+458) km$^2$, 2001 – 2320 km$^2$, 2018 – 1331 km$^2$, background – 1109 km$^2$).
Fig. 2b (Fig. 8b in the new version of the manuscript) shows the distributions of NDVI for the burned areas detected in 1968+1988, 2001, 2018, and for background areas. Using the dates of major fires, we can assume that 1968+1988 data show the state of vegetation in the site burned more than 42 years ago, 2001 data – 28 years ago, 2018 data – 2 years ago and background data refer to tundra not affected by fires during the whole study period.

The distributions are close to Gaussian ones for the background tundra and recently burned sites. Interestingly, the distributions are bimodal in the sites burned 28 and >42 years ago and they moved to higher NDVI values as compared to the background site. We fitted the distributions by the sums of two Gaussian functions and determined mean values and standard deviations for all the peaks (Table 1 in this document, Table 7 in the new version of manuscript). We found that bimodal distributions had almost the same two peaks. However, in the distribution from the sites burned 28 years ago, the peak with lower NDVI was more pronounced as compared to the peak with higher NDVI. Oppositely, in the sites burned >42 years ago, the peak with higher NDVI became more pronounced.

[Figure]

Fig. 2. (a) The distribution of NDVI over study area 1 in 2018. The curves of different colors indicate boundaries of the background tundra sites and burned tundra sites detected in the Corona and Landsat mosaics from different years (see legend). Burned areas in the mosaics from 1968 and 1988 are mainly due to fires from >42 years ago, in 2001 – due to fires from 28 years ago, in 2018 – due to fires from 2 years ago. (b) Distributions of NDVI based on the data from background sites and the sites burned in different years.

Table 1. Parameters of fits of the NDVI distributions in Fig. 2b by Gaussian functions

$$N_{pix} = A_1 \exp\left(-\frac{(NDVI - NDVI_{max,1})^2}{2\sigma_1^2}\right) + A_2 \exp\left(-\frac{(NDVI - NDVI_{max,2})^2}{2\sigma_2^2}\right).$$

| Year | $A_1$ | $NDVI_{max1}$ | $\sigma_1$ | $A_2$ | $\sigma_{max2}$ | $stdv_2$ |
|---|---|---|---|---|---|---|
| 1968+1988 | 489 | 0.6557 | 0.0741 | 496 | 0.7762 | 0.0393 |
| 2001 | 1459 | 0.6251 | 0.0472 | 598 | 0.7493 | 0.0566 |
| 2018 | 366 | 0.5385 | 0.0890 | 554 | 0.5471 | 0.0463 |
| Background | 853 | 0.5934 | 0.056 | - | - | - |

Further, we used the mean values and standard deviations to identify vegetation associated with the peaks of the distributions in the satellite images. For illustration, we chose an image containing all representative examples of vegetation (Fig. 3 in this document, Fig. 9 in the new version of manuscript).

Green color in Fig 3, right panel, indicates the pixels which have NDVI in the interval ($NDVI_{max,2}$-$\sigma_2$; $NDVI_{max,2}$+$\sigma_2$) corresponding to the upper peak of the distribution based on the data from

1968+1988. This peak is mainly associated with forests. The lower peak (pixels in blue color, Fig. 3b), as can be seen from Fig. 3, corresponds to woodlands and tundra. This lower peak has a large intersection with the peaks in the distributions based on the data after recent fire and background data. However, interestingly, there is a significant decrease in the pixels with NDVI below ca 0.52 in the bimodal distributions. These sites are marked in pink in Fig. 3, right panel. They correspond to the tundra sites lightest in color due to the presence of lichen in the vegetation community. The fraction of such pixels decreases in bimodal distributions, meaning that lichen does not recover to previous state. Instead, bimodal distributions gain a large fraction of pixels with high NDVI corresponding to forests, absent in background tundra.

[Figure]

- woodland/tundra, 1968+1988;    - background tundra;    - burned tundra, 2018

- forest, 1968+1988;    - background tundra, low NDVI;

Fig. 3. Representative types of vegetation associated with different state of the sites and NDVI. Left panel: an image without mask, right panel: the same image colored according to the state of the site (burned in mosaics from 1968+1988 or 2018, background) and NDVI. In the right panel, green color corresponds to the upper peak and blue color corresponds to the lower peak in the bimodal distribution from the sites burned in 1968/1988 (Fig. 2b). Pink color marks pixels with NDVI lower than 0.52 in the background site.

We compared NDVI distributions based on Landsat 8 data from 30 June 2018 and 03 July 2019 that we used in the previous version of the manuscript (Fig. 4 in the current document). The distributions of NDVI are similar for both years, but in 2019, the upper peak in the bimodal distributions is less pronounced. This could be due to different phenological states of vegetation, dependent on temperature and precipitation from year to year. The peaks corresponding to tundra vegetation and woodlands almost did not change their position in both figures, but the peak of the distribution after the recent fire (2018 in the legends) and the forest peaks in bimodal distributions have larger mean NDVI in 2018.

Finally, we estimated the fraction of forest in bimodal distributions. We used NDVI data from 2018, as the separate peaks were better pronounced in bimodal distributions. Using standard deviations of the two peaks, the boundary approximately separating forest peak from tundra peak corresponds to the threshold value of NDVI=0.73. We assume that pixels with NDVI > 0.73 represent mainly forest, and pixels with NDVI<0.73 – tundra and woodlands. We integrated the distribution to find the fraction of pixels with NDVI>0.73 in the total number of pixels. In the areas burned 28 years

ago, forests occupied 24% of the total area. In the areas burned more than 42 years ago, the forest fraction increased to 55% of the total area. This number is comparable to our estimates of the vegetation shift within forest-tundra zone (56%) and exceeds the estimates for the northern taiga zone (14%).

[Figure]

Fig. 4. Distributions of NDVI in the burned and background areas based on the images from 30 June 2018 (left panel) and 3 July 2019 (right panel).

While precise estimates of forest fraction based on NDVI are challenging, the main results following from the distributions in Fig. 2b can be summarized as follows:

1.  The NDVI distributions based on the data from background tundra and areas burned 2 years ago are predominantly unimodal, whereas the distributions based on the data from areas burned 28 years ago and earlier are bimodal.
2.  The low NDVI pixels corresponding to vegetation communities in tundra characterized by relatively high amounts of lichen and thus having lightest colors in the images largely disappear from the distributions corresponding to vegetation communities recovered after fires.
3.  Instead, the new state of vegetation recovered after fires is characterized by a higher mean NDVI due to a new peak associated with forest. The fraction of pixels representing forest increases with time after the fire.

We thank again the referee for the useful suggestions. We hope that the present manuscript addresses all the comments raised.

---

## Author Response (AR1)

**Replies to the comments of Referee 1**

We are grateful to the referee for the constructive criticism, which helped to improve the clarity of the manuscript. Please find below the replies to the comments and an account of the modifications implemented.

**General comments**

1. *The objectives as formulated in the introduction are not followed by a corresponding structure and sequence in the methods and results section. This makes the overall manuscript very hard to follow as a reader, as one needs to search for the corresponding information.*

We apologize for this inconvenience. We have now organized the structure of the manuscript (in Result and Discussion sections) following the objectives in the introduction (Lines 56-62). First, we calculate climatic indices to show the possibility of vegetation shift; second, we study dynamics of fires; third, we study dynamics of vegetation and its links to fires and topography.
"The main objectives of the study are:
1) to study dynamics of regional climatic factors in order to assess the possibility of vegetation shifts due to climate change, in particular tundra-forest transition;
2) to quantify burned surface areas and calculate frequency of wildfires;
3) to study the link between wildfires and dry tundra transition to woodlands and forests.
Finally, we take into account physiographic characteristics of the landscape and study the link between the topographic slope and the transition."

The structure of the manuscript is now organized as follows:
1 Introduction
2 Materials and methods
 2.1 Field sites
   2.1.1 General description
   2.1.2 In-situ observations of vegetation and permafrost state
 2.2 Calculation of climatic indices
 2.3 Wildfires
 2.4 Vegetation dynamics
 2.5 Topographic slopes
3 Results
 3.1 Temperature, precipitation and climatic indices
 3.2 Dynamics of fires
 3.3 Vegetation dynamics and its link to fires and topography
   3.3.1 Assessment of vegetation recovery after fire using NDVI
   3.3.2 Assessment of vegetation shift using visual method, connection to fires and topography
4 Discussion
5 Conclusion

2. *For example, there is not a dedicated methods section that explains how the first objective (to quantify burned surface areas and assess frequency and causes of wildfires) was addressed and the sequence is changing between methods and results. Apart from structural problems, some of the objectives are not directly followed at all or in a qualitative way only*

We have described how we quantified burned areas (please find in the answer to the next question) and frequency of wildfires in Methods, section 2.3 Wildfires (Lines 135-154).

'We studied the percentage and distribution of the burned sites and calculated the frequency of fire return. Corona and Landsat images showed that some years were characterized by particularly large-scale fires in the study areas (see an example for 1990 in SM, Fig. SB1). These years are referred to as the years of major fires. The burned areas can be detected in the satellite images during a few years after the fire. Landsat mosaics from 1988, 2001, 2016 and 2018 largely reflect the state of the study areas after the major fire years 1976, 1990, 2012 and 2016 (Table 5). Burned areas in Corona mosaic from 1968 were partially dated back to the fires in the period 1953-1964 based on geological surveys and early Corona images (the sources are listed below Table 5). For a given territory, the period between fires was calculated as the difference in years between the major fires.'

In order to avoid qualitative results, we decided not to address causes of wildfires in the current manuscript and leave this topic for a more careful quantitative study in future. We have removed all the information pertaining to possible causes of fires from the manuscript. We have also removed section 'Qualitative observations of the vegetation dynamics'.

> 3. *Methods: The manuscript contains tables with data sets, but it remains unclear which data sets were used for which objectives/results specifically, how the imagery was preprocessed given so many different data sets of highly varying spectral, spatial resolution and quality were used. Also, details on the processing of data (esp. remote sensing data, e.g. atmospheric correction) are largely missing (indicating a software without even the version or parameters used for the algorithm is not sufficient for reproducible methods). Further, there is little to no information about validation of the classification results or reference to uncertainties of results.*

We have added the information about data sets and preprocessing of the data in Methods (sections 2.3 Wildfires and 2.4 Vegetation dynamics).

Wildfires
*Lines 124-128*: 'The initial state of the study areas was assessed using Corona images. We identified 21 frames under clear-sky conditions from 21 August 1968. Each frame consisted of 4 scanned fragments. The cropped fragments without color correction were georeferenced to the chosen orthomosaic (SPOT layer, section 2.4) using polynomial method (3rd order) in software ArcGIS (v. 10.4.1). The r.m.s. error of georegistration estimated using control points did not exceed 10-12 m. Then the fragments were organized in separate paths and finally a mosaic was formed using mosaic operator Last in ArcGIS.'
*Lines 130-134:* 'Further, we used Landsat data to quantify burned areas. The data providing the best coverage of the study areas were available from the following years: 1988, 2001, 2016 and 2018 (Table 4). The images were synthesized using near- and mid-infrared channels (Landsat 5 and 7: 0.63-0.69 µm, 0.76-0.90 µm and 1.55-1.75 µm; Landsat 8: 0.64-0.67 µm, 0.85-0.88 µm and 1.57-1.65 µm) as burned areas are visible in the infrared range of wavelengths. Landsat mosaics for all years were formed with application of color correction using mosaic operator Blend in ArcGIS.'
*Lines 135-147*: 'Mapping and quantification of the burned areas were performed by means of an object-based image analysis, successfully used for studies of landscape dynamics (Blaschke, 2010). On the first stage, we performed segmentation of mosaics using algorithm 'Multiresolution segmentation' in software eCognition (v. 9.0). The segmentation was done using parameters 40 for Scale and 0.5 for Color. The second stage, classification, was different for Corona and Landsat

mosaics. For Landsat mosaics, we used unsupervised classification ISODATA (15 classes, a change threshold 5%). Further, we identified visually one or two classes corresponding to burned areas. The segments containing more than 90% of pixels within these classes were identified as burned areas. In addition, we checked the segments with lower percentage of pixels (down to 40-50%) belonging to these classes and they were manually added to burned areas when necessary. In Corona mosaics, the spectral information was missing and we had to rely on the contrast of colors between background tundra and burned areas. For visual classification, we applied two criteria. First, background tundra is lighter due to the presence of lichen in vegetation community, whereas recently burned areas are dark. Second, burned areas are characterized by well-defined boundaries often coinciding with river coastlines. An example illustrating segmentation and visual choice of burned areas is shown in Fig 3. Calculation of the areas of segments classified as burned areas was performed using standard instrument Calculate geometry in ArcGIS.'

Note that we tried both supervised and unsupervised classifications, but all methods were semi-automatic due to the high inhomogeneity of the region. For a number of segments, the AI-methods helped to identify potential burned areas whereas an observer had to make final decision (using Normalized Burn Ratio and/or visual inspection). We did not add ambiguous segments, thus our method likely gives a lower estimate of the burned areas.

Vegetation dynamics
*Lines 160-162:* 'The initial state of vegetation was obtained from Corona imagery and topographic maps. The compilation of mosaic using Corona images is described in Section 2.3. The topographic maps were used to develop forest and dry tundra masks using automatic tracing in software EasyTrace (v. 8.7). The resulting vector layer was checked and corrected using Corona mosaic.'
*Lines 163-167:* Assessment of the current state of vegetation was based on Resurs-P and SPOT data (Table 6). Majority of the territory was covered by mosaic of SPOT-6,7 imagery synthesized in the visible range without color correction. Ca 10% of the territory were covered by three paths of Resurs-P (the product level 2A, including four channels B, G, R and NIR). The SPOT mosaic was used as a pluggable webmap layer without additional processing. We co-registered Resurs-P data to the SPOT mosaic in ArcGIS.'

Dynamics of vegetation was assessed using the visual method described in Section 2.4 'Vegetation dynamics'. Overall, visual methods are not rare in scientific studies. For example, classification of clouds performed by observer is typically taken as an etalon when automatic methods are developed. It is also not the first time when the visual methods are used for quantification of the vegetation shift. In the pioneering study, Frost and Epstein (Global Change Biology (2014) 20, 1264–1277) used a similar visual method for the same purpose. They state that 'Gambit and Corona are well suited for land-cover change studies in tundra ecotones because tall shrubs and trees form abrupt transitions in vegetation structure that create unambiguous, readily interpreted photo-signatures. These photo-signatures result from the shadowing projected by the canopies of tall shrubs and trees, which greatly overtop tundra vegetation and create areas of high contrast in panchromatic imagery.'
We have added a figure illustrating different decisions about the vegetation shift (Fig. 4 in the manuscript) and added reference to Frost and Epstein (2014) study in the methods (Lines 158-159). We supported visual analysis of vegetation dynamics by the analysis based on NDVI (see the answer to p.4 of Specific comments).

4. *Overall the presentation of the manuscript is really not sufficient – as indicated in more detail below, graphs are poor (missing legends (esp. Fig 1 & 2), scale, missing reference of Figure in main text). Consider a more rigorous selection of graphs and information*

*displayed in tables. Also, thorough revision of language (esp. articles) and checking of consistency are needed to make this manuscript more accessible.*

We would like to thank reviewer for valuable comments regarding the figures and tables of the manuscript. Here is the list of modifications/addings done in the figures (the numbers correspond to the previous version of the manuscript):

Fig. 1: we added the legend. We added boundaries 'southern tundra - forest-tundra –northern taiga' in the figure and extended the figure caption.

Fig. 2: we removed the figure.

Fig. 3 (Fig 2 in the current version): we extended the figure caption.

Fig. 4a (Fig. 5a in the current version): we added linear trends of temperature as requested by the reviewer and extended the figure caption.

Fig. 5b (Fig. 6b in the current version): we marked years of major fires by dashed lines to emphasize connection between fires and evapotranspiration.

Fig. 6 was erroneously referenced as Fig. 5 in the previous version of the manuscript. In the present version, we have removed the section where this figure should have been referenced, as well as the figure.

Fig. 8a: we marked dry tundra in the NDVI mosaic and added 'background' in the legend.

Fig. 8b with mean NDVI is replaced by the figure with NDVI distributions.

Fig. 9 (Fig. 10 in the current version): we retain only panel (a) and added boundaries 'southern tundra - forest-tundra –northern taiga' in the figure.

Fig. 11(Fig. 12 in the current version): only the figures with the mean slopes are retained.

In the current version, we have added the following new figures:

Fig. 3: illustrates segmentation and classification in Corona mosaics,

Fig. 4: illustrates decision about vegetation change using visual method,

Fig. 9: illustrates types of vegetation corresponding to pixels with different NDVI.

From the tables, we have removed the information related to causes of fires and classification of vegetation zones based on recent 'Atlas…', 2004 (Tables 1 and 4 in the current version, Tables 2 and 3 in the previous version).

We have made revision of language throughout the text and reorganized paragraphs in several sections to make them more consistent (e.g., In-situ observations of vegetation and permafrost state, Discussion). The situation with articles will be further improved if the manuscript is accepted, because all EGU journals including BG support English correction before the manuscript is published.

**Specific comments**

1. *Temp., precip, climatic indices - Methods for GDD5 (line 144) – provide reference for this formula.*

We added the reference: 'We calculated growing degree-days following Tchebakova et al. (1994)' (Line 113).

*Line 158 – what is the 3∘ increase based on – a trend fitted to the climate data in Fig. 4? If yes, show the trend and related statistical information.*

Yes, and we added the trends and the corresponding statistical information in Fig. 4a (Fig. 5a in the current version).

> *- Data from 3 meteorological stations are presented. It remains unclear how these are linked to the 3 selected study sites as the stations are located outside of the study sites and not in obvious pairing to the 3 sites.*

These three stations cover the whole range of latitudes of our study areas and they are located reasonably close to the study areas. One of them (Nyda) is located within study area 1, but its latitude is also close to that of the northern part of study area 2. Two stations (Novy Port and Nadym) are indeed located outside the study areas. Station Nadym is located near the southern borders of two study areas, between areas 1 and 3 (50 km from the study area 3, 150 km from the study area 1). It should be representative of the climatic conditions in the southern parts of study areas. Station Novy Port is located 70 km to the north of study area 1 and this is the closest station to the northern boundaries of the study areas. The latitudes of the northern part of study area 1 are between those of stations Nyda and Novy Port.

Growing degree-days, an index based on air temperature (2 m height), should be largely the same along the latitude. Precipitation can significantly vary from station to station, which is also seen in our current Fig. 6. However, the dryness index is not a limiting parameter for the vegetation shift at any of the stations (at least according to the model SibCLIM) and we do not expect this parameter to prevent vegetation change anywhere within our study areas.

We added the information about location of stations in the manuscript (Lines 103-106): 'Station Nadym (65°32'N, 72°32'E, 7 m a.s.l.) is located near the southern borders of the study areas (50 km from study area 3). Station Nyda (66°37'N, 72°57'E, 10 m a.s.l.) is located within study area 1. Station Novy Port (67°41'N, 72°52'E, 12 m a.s.l.) is located to the north of the study areas. This is the closest station to the northern boundaries of the study areas.'

> *For example on line 180 it is stated that based on the climate data analysis, the vegetation class in Novy Port changed from forest-tundra to dark needled northern taiga - how are the climate data linked to vegetation classes, and the station data to the study sites?*

The vegetation classes obtained from the topographic maps do not follow the classification based on the climatic indices. For example, at the latitude of Nyda, the map shows southern tundra, while the climate-based classification suggests dark needed northern taiga. This is likely the consequence of the insufficient precision of vegetation classification based on climatic indices, which can be too rough in the transitional zones. Climate data analysis is used in our study to show that theoretically conditions over all the study areas are suitable for forests.

> 2. *Qualitative assessment of vegetation dynamics. Overall this section is not convincing as it is largely missing a corresponding reproducible methods section. Quantitative results are hard to reach based on Corona as reference data set. But even if only qualitatively assessed, methods need to be clearly outlined.*

We have removed this section.

> *- How were these transitions qualitatively assessed? Some information can be found in section 2.1, some in the field sites general description, but nowhere is clearly formulated how the transitions were visually/qualitatively assessed, what classes were followed.*

We added Fig. 4 in the current manuscript to illustrate how vegetation change was assessed.

> *Also, it remains unclear how the topographic map was used for this (does it contain forested area? Burned area?).*

It contains forested areas but not burned areas and we used it to create our forest mask. We added the following information in the Section 2.4 Vegetation dynamics (Lines 161-162):
'The topographic maps were used mainly to develop the forest and dry tundra masks using automatic tracing in software EasyTrace (v. 8.7). The resulting vector layer was checked and corrected using Corona mosaic.'

> *The graphs that are mentioned to highlight how this was done are not conclusive (e.g. Fig2 misses a color legend, also it is not clear from this graph which of the layers show the most reliable forest cover.*

> *Fig SB3 – without clear indication in the imagery it is hard to understand where the active afforestation mentioned in the figure title is located – this is certainly due to the very different quality of the Corona versus Yandex map layers, but as presented does not convince the reader that this active afforestation has happened). Also, how many sites (burned and background sites) were assessed in total? Are the different conditions statistically balanced (for several of the assessed transitions only a single reference site is mentioned, does it mean that this condition was only observed once)? What was the exact sampling design? - What is active afforestation? Define in the related methods section - L. 191- fig 5 is wrongly referenced (Fig 5 displays potential evaporation, not tundra after wildfire) - L 204 – removal of vegetation cover – define in related methods section.*

Certainly, the sites were not statistically balanced as some conditions are quite specific (e.g., river valleys) but the conclusion was never made based on a single reference site. We have removed this section and this part of Supplementary material.

> 3. *Dynamics of fires - Methods: classification to identify burned areas: how was the initial state determined in the Corona images?*

Added in section 2.3 Wildfires (Lines 142-147), see also answer to Q3 of General comments.

> *In table 2 you also list Sentinel, Modis and VIIRS data – how were these data used (you only mention Corona and Landsat in the fire methods section)*

We used these data to study qualitatively hot spots of the fires, but we removed this information from the current version of the manuscript.

> 4. *Dynamics of vegetation and fires - NDVI is NOT the normalized digital vegetation index*

We apologize for this error, which was unfortunately copied several times. We changed 'digital' to 'difference' throughout the text.

> *- Which imagery was used to calculate NDVI? Any preprocessing performed? Georegistration issues discovered? Explanations on remote sensing data in methods are insufficient. –*

We added this information in Section 2.4, Vegetation dynamics (Lines 187-189):

'NDVI was calculated based on two scenes from one path of Landsat-8 from 30 June 2018 (path/row 160/013 and 160/014).We used Level 2 data (CEOS) after atmospheric correction by the standard Landsat 8 OLI atmospheric correction algorithm (Vermote et al., 2016). NDVI was calculated in ArcGIS using standard tools'.

For ArcticDEM and satellite data sets except Corona, georegistration was performed by the data owners during orthotransformation of the satellite images. For topographic maps, georegistration was performed using coordinates of the check points in the field. The rms errors of georegistration for different types of data are reported below:
Resurs-P 1,2 – 5-10 m,
SPOT-6,7 – 5-8 m,
ArcticDEM – 5-7 m,
topographic maps – 20-25 m,
Landsat – below 15 m,
Corona/KH-4B – 10-15 m.

Corona images were georegistered to the mosaic of SPOT images in ArcGIS using polynomial method (3$^{rd}$ order). According to the data provider (CNES, National Centre for Space Studies, France), for SPOT images, the rms of georegistration is 5-8 m and this was confirmed in different studies (e.g. Parage et al. New sensors benchmark report on SPOT 7, 2014).
For each of 88 frames of 'Corona', there is a number of check points (see figure below) for which we estimated rms errors along the latitude (NS), along the longitude (EW) and 2D rms error. For all the frames, the rms error was below 10-15 m.

[Figure]

Fig. 1. Georegistration of Corona frames to the SPOT mosaic

*Fig 8 – what is displayed here exactly? This remains unclear based on the corresponding methods section and figure title. Is the standard deviation based on spatial variation for the background sites? How is temporal variation in NDVI of the background sites accounted for? Are the different years and the background areas statistically balanced for their size?*

In the previous version of the manuscript, Fig. 8a showed the distribution of NDVI over study area 1 based on the Landsat data from 3 July 2019. The curves of different colors indicated boundaries of the background tundra sites and burned tundra sites detected in the Landsat mosaics from different years (see legend). Fig. 8b showed mean NDVI indices and standard deviations calculated

for the background areas and the areas burned in different years as indicated in panel (a). The standard deviation was based on the spatial variation of NDVI (determined by the number of pixels with different NDVI within each area) and temporal dynamics was not accounted for. Instead of mean values and standard deviations, in the current version of the manuscript we use distributions of NDVI and extend NDVI-related analysis (sec. 3.3.1). We also show NDVI based on Landsat data from 30 June 2018 instead of data from 3 July 2019 for the reason described below. Therefore, Fig. 8 has changed (see Fig. 2 in this document).

In order to make the areas more balanced by size, we merged the data sets for 1968 and 1988, for which we might expect that vegetation has recovered after fires, based on the mean NDVI values. As a result, all study areas were larger than 600 km$^2$ (1968+1988 – 600 (405+195) km$^2$, 2001 – 1565 km$^2$, 2018 – 867 km$^2$, background – 937 km$^2$). (*Lines 193-195*)

Fig. 2b (Fig. 8b in the new version of the manuscript) shows the distributions of NDVI for the burned areas detected in 1968+1988, 2001, 2018, and for background areas. Using the dates of major fires and the date of Landsat mosaic (2018), we can assume that NDVI within burned tundra sites detected in 1968 and 1988 reflect the state of vegetation after fires more than 42 years ago, in 2001 – after fires 28 years ago, in 2018 – after fires 2 years ago. NDVI in background tundra refers to tundra not affected by fires during the whole study period.

[Figure]

Fig. 2. (a) The distribution of NDVI over study area 1 on 30 June 2018. Segments with boundaries of different color are background dry tundra sites and burned dry tundra sites detected in the Corona and Landsat mosaics from different years (see legend). Burned areas in the mosaics from 1968 and 1988 are mainly due to fires from >42 years ago, in 2001 – due to fires from 28 years ago, in 2018 – due to fires from 2 years ago. (b) Distributions of NDVI based on the data from background sites and the sites burned in different years. In the legend, numbers in brackets indicate years after the last major fire.

The distributions from the background tundra and recently burned sites are close to Gaussian ones. Interestingly, the distributions from the sites burned 28 and >42 years ago are bimodal and they have higher NDVI values as compared to the background and recently burned sites. We fitted the bimodal distributions by the sums of two Gaussian functions and determined mean values and standard deviations for all the peaks (Table 1 in this document, Table 7 in the new version of manuscript). The positions of the lower peaks of the bimodal distributions differ only slightly, whereas the position of the upper peak is a bit lower and the peak is less pronounced for the area burned 28 years ago.

Further, we used the mean values and standard deviations to identify vegetation associated with the peaks of the distributions in the satellite images. For illustration, we chose an image containing all representative examples of vegetation (Fig. 3 in this document, Fig. 9 in the new version of manuscript).

Green color in Fig 3, right panel, indicates the pixels which have NDVI in the interval (NDVI$_{max,2}$-$\sigma_2$; NDVI$_{max,2}$+$\sigma_2$) corresponding to the upper peak of the distribution based on the data from 1968+1988. Comparing left panel and right panel of Fig. 3, one can conclude that this peak is mainly associated with forests. The lower peak (pixels in blue color in Fig. 3, right panel), as can be seen from Fig. 3, corresponds to woodlands and tundra. This lower peak has a large intersection with the peak in the unimodal distribution from the background site (Fig. 2b). However, interestingly, there is a significant decrease in the pixels with NDVI below ca 0.52 in the bimodal distributions. These pixels are marked in pink in Fig. 3, right panel. They correspond to the tundra sites lightest in color due to the presence of lichen in the vegetation community. The fraction of such pixels decreases in bimodal distributions, meaning that lichen does not recover to its state before the fire. Instead, bimodal distributions gain a large fraction of pixels with high NDVI corresponding to forests.

Table 1. Parameters of fits of the NDVI distributions in Fig. 2b by Gaussian functions

$$N_{pix} = A_1 \exp\left(-\frac{(NDVI - NDVI_{max,1})^2}{2\sigma_1^2}\right) + A_2 \exp\left(-\frac{(NDVI - NDVI_{max,2})^2}{2\sigma_2^2}\right).$$

| Year | A$_1$ | NDVI$_{max1}$ | $\sigma_1$ | A$_2$ | NDVI$_{max2}$ | $\sigma_{max2}$ |
|------|-------|---------------|------------|-------|---------------|-----------------|
| 1968+1988 | 308 | 0.64 | 0.07 | 132 | 0.77 | 0.04 |
| 2001 | 954 | 0.62 | 0.04 | 448 | 0.71 | 0.06 |
| 2018 | 552 | 0.54 | 0.07 | - | - | - |
| Background | 726 | 0.59 | 0.06 | - | - | - |

[Figure]

■ - woodland/tundra, 1968+1988;  □ - background tundra;  ▨ - burned tundra, 2018

■ - forest, 1968+1988;  ■ - background tundra, low NDVI;

Fig. 3. Representative types of vegetation associated with different state of the sites and NDVI. Left panel: an image without mask, right panel: the same image colored according to the state of the site (burned in mosaics from 1968+1988 or 2018, background) and NDVI. In the right panel, green color corresponds to the

upper peak and blue color corresponds to the lower peak in the bimodal distribution from the sites burned before 1968/1988 (Fig. 2b). Pink color marks pixels with NDVI lower than 0.52 in the background site.

We compared NDVI distributions based on Landsat 8 data from 30 June 2018 and 03 July 2019 (Fig. 4 in the current document). These dates were chosen close to the peak of daily temperature and should be representative of the peak growing season. The distributions of NDVI are largely similar for both years. However, for NDVI based on mosaic from 2019, the upper peak in the bimodal distributions is less pronounced (in 2001-distribution it looks more like asymmetry) and there appears a lower peak in the distribution corresponding to recent fires. This could be due to different phenological state of vegetation, dependent on temperature and precipitation from year to year. The peak corresponding to tundra vegetation and woodlands almost did not change its position in both figures, but the peak of the distribution after the recent fire (2018 in the legends) and 'forest' peaks in bimodal distributions have larger NDVI in 2018.

Finally, we estimated the fraction of 'forest' pixels in the bimodal distributions. We used NDVI data from 2018, as the separate peaks were better pronounced in bimodal distributions. Using standard deviations of the two peaks, the boundary approximately separating forest peak from tundra peak corresponds to the threshold value of NDVI=0.72. We assume that pixels with NDVI > 0.72 represent mainly forest, and pixels with NDVI<0.72 – tundra and woodlands. We integrated the distribution to find the fraction of pixels with NDVI>0.72 in the total number of pixels. In the areas burned 28 years ago, forests occupied 19% of the total area. In the areas burned more than 42 years ago, the forest fraction increased to 28% of the total area. This number is comparable to our estimates of the vegetation shift within the forest-tundra and northern taiga zone (14-56%).

[Figure]

Fig. 4. Distributions of NDVI in the burned and background dry tundra areas based on the images from 30 June 2018 (left panel) and 3 July 2019 (right panel). In the legend, years correspond to these of mosaics used to calculate burned territories, numbers in brackets indicate years after last major fire.

While precise estimates of forest fraction based on NDVI are challenging, the main results following from Figs. 2 and 3 can be summarized as follows:
1. The NDVI distributions based on the data from background tundra and areas burned 2 years ago are predominantly unimodal, whereas the distributions based on the data from areas burned 28 years ago and earlier are bimodal.
2. The low-NDVI pixels corresponding to vegetation communities in tundra characterized by relatively high amounts of lichen and thus having lightest colors in the images largely disappear from the distributions corresponding to vegetation communities recovered after fires.

3. Instead, the new state of vegetation recovered after fires is characterized by a higher mean NDVI due to the new peak associated with forest. The fraction of pixels representing forest increases with time after the last fire.

We thank again the referee for the useful suggestions. We hope that the present manuscript addresses all the comments raised.

**Replies to the comments of Referee 2**

Please find below the replies to the comments.

1. *I also appreciate the involvement of the Corona satellite, which allows for extending the observation period further back in time compared to Landsat, although I wonder how the authors come to a period of 60 years (2018-1968=50).*

Burned sites detected in 'Corona' imagery from 1968 can be dated back to fires from the period between 1953 and 1964 based on geological surveys and separate images from early Corona mission (Table 6 in the previous version of the manuscript, Table 5 in the current version). We added Lines 151-153 in the current version of the manuscript: ´Burned areas in Corona mosaic from 1968 were partially dated back to the fires in the period 1953-1964 based on geological surveys (Chekunova V.S., Geological and geomorfological survey of a part of the lower reaches of the River Nadym basin and parts of the right bank of the River Ob in Nadym region in 1953. VSEGEI: 1954. 72 p.) and early Corona images`. Therefore, the starting point of our study is in fact between 54 and 65 years ago, which is on average closer to 60 years than to 50.

2. *Nevertheless, I agree with reviewer 1 in all the points made: in its current shape the manuscript contains too many unknowns to be able to make a proper judgement of the results. I got strong feeling that many of the results (e.g. classification was based on visual interpretation).*

We do not fully agree with the statement that the manuscript contains too many unknowns. We added the required information in section Methods and described it in the replies to reviewer 1. The manuscript contained many quantitative results already in its first version (version with what both reviewers were working), e.g., quantification of burned areas, period between fires, recovery times based on NDVI, change of the vegetation state. However, in the current version of the manuscript, we aimed to keep only quantitative results and we have removed all qualitative results (the information about possible causes of fires and section 'Qualitative observations of the vegetation dynamics').

3. *First the major issues highlighted by reviewer 1 need to be addressed before I can provide an in-depth review of the manuscript.*

We have addressed the issues raised by reviewer 1. The point-to-point answer to reviewer 1 and the manuscript with additions marked in blue color have been uploaded to the system. We hope that the reviewer will now be able to provide an in-depth review of the manuscript.

[revised manuscript text omitted]

**Supplementary material.**
**Section A. Dynamics of climatic variables and indices.**

**Meteorological stations:**
**Nadym** (65°32'N, 72°32'E, 7 m a.s.l.)**, Nyda** (66°37'N, 72°57'E, 10 m a.s.l.)**, Novy Port** (67°41'N, 72°52'E, 12 m a.s.l.)

[Figure]

Fig. SA1. Seasonal cycles of monthly temperature and precipitation over 1966-2010. Red risks denote median values, bottom and top edges of boxes are 25th and 75th percentiles.

[Figure]

Fig. SA2. Long term-trends, Novy Port: length of growing season (a), mean temperature of growing season (b) minimum daily temperature (d), maximum daily temperature (e); correlation of growing degree-days with the length of growing season (c), color scale – mean temperature of growing season.

[Figure]

Fig. SA3. Same as SA2, for Nyda.

[Figure]

Fig. SA4. Same as SA2, for Nadym.

[Figure]

Fig. SA5. Dryness index, the ratio between potential evaporation and precipitation.

**Section B. Satellite and topographic maps**

[Figure]

Fig. SB1. Illustration of the major fires of 1990 in Landsat 5 mosaics from 1990/1991. A – before fires (15 June - 12 July 1990), B – active fires (17, 18, 28 July 1990), C – after fires (22 July - 30 August 1991). © Microsoft.

[Figure]

Fig. SB2. Coverage of the study areas by separate sheets of the topographic map.

---

## Author Response (AR2)

**Replies to the comments of referee 2**

*The authors have done a good job in addressing many of the previously raised concerns and the paper is much more easily accessible now. However, there are still a few remaining issues on the content, and many details to address. Also a thorough language check is needed.*

We are grateful to the referee for the positive estimate of our work. The language has been checked by two co-authors, we hope it has become better. Please find the answers to the specific comments below.

*Major points*
*- research question 1 and 3 are based on different vegetation definitions (section 1 on definitions as shown in table 5, section 3 uses the terms tundra, woodlands, and forests, without defining them further. Are these the same categories? For the consistency of the narrative of the paper, it is important that categories and definitions are precise and consistent across all sections. If the definitions in the 2 sections are not the same, then it is questionable how these two sections are related and part of the same paper.*

We thank the referee for this comment. In the manuscript, we have vegetation classes (or zones) and vegetation types. The terms 'tundra' and 'taiga' (forest) can be used for both classes and types. Vegetation classes are zones where a certain type of vegetation prevails. Within one zone there can be different vegetation types. For example, there can be tundra as a vegetation type in the northern taiga zone. Types and classes (zones) are now used consistently throughout the manuscript.

In Sec. 1 (Table 3), we distinguish between the following vegetation classes: Tundra, Spruce-larix forest-tundra, Dark-needled northern taiga, and Dark-needled middle taiga. Similarly, the following classes were used in the topographic maps: southern tundra, forest-tundra, and northern taiga (Figs. 1, 9). In addition, we used topographic maps to study the distribution of vegetation types within a zone.

Our analysis in Sec. 1 suggests that the boundaries of the zones have changed, and our study areas are now located within class or zone 'taiga'. If the zone boundaries have moved, the vegetation types should have changed too. In Sec. 3, we study transition of vegetation types within different zones, as defined by the topographic map. We use the following vegetation types: tundra (more precisely, dry tundra), woodlands (<50% of the area occupied by trees) and forests (>50% of the area occupied by trees). These types of vegetation were defined in Methods, Section 'Vegetation dynamics', Lines 187-188.

*Minor issues*
*- sort all sections logically, e.g. consistently from north to south when describing vegetation and climatic stations, and early to later dates (e.g. first Corona image description, then Landsat)*

We have sorted the sections as suggested: e.g., from north to south when describing stations in sec. 2.2 and 3.1, Corona before Landsat in sec. 2.3 (description of classification). In addition, we changed the structure of sec. 2.4 (first, NDVI-based method, then visual method) to follow the structure of sec. 3.3.

*- for all satellite data: state source/reference of data (where downloaded), product level (DN, calibrated radiance, surface reflectance product). Also, you need to justify if no atmospheric correction was performed.*

The sources were listed under Tables 3-5. Product levels are reported in Methods (Level 1 data for fire analysis (Landsat), Level 2 data for NDVI analysis (Landsat), Level 1C data (Resurs-P)), see L 125, 160, 179. For fire analysis data we did not perform atmospheric correction because geometric correction is sufficient for the calculation of area (Song et al., Classification and change detection using Lansat TM data: when and how to correct atmospheric effects? Remote sensing of environment (2001) 75:230-244). Working with NDVI, we used the data after atmospheric correction (L 160-161).

*l2: replace plant species by vegetation types*

Changed as recommended.

*L2: what kind of models are meant here? specify*

We mean bioclimatic models, added in L 3.

*L13: remove 'data set from'*

Changed as recommended (L 14).

*L14: remove 'in the Arctic as a whole' – duplication*

Changed as recommended (L 15).

*L22: Shrubification and afforestation in the boreal zone is not expected to provide a means for climate mitigation as the positive feedback to warming through albedo decrease is higher than the relatively low additional carbon sequestration and storage by trees. Revise.*

We do not fully agree. We have just published a study providing estimates of carbon balance change for grassland-forest transition in the boreal zone accounting for albedo effect and atmospheric feedbacks (Kulmala, Ezhova, Kalliokoski et al. CarbonSink+ - Accounting for multiple climate feedbacks from forests, 2020: Boreal Env. Res. 25: 145–159. Publication date: 11 November 2020). There we show that surface albedo effect can be 1) strongly reduced due to dense cloud cover and short daytime in winter; 2) compensated by atmospheric feedbacks.
Albedo effect was mentioned in conclusions but we have now added it in the introduction too. L20-23: 'An increase in warm degree-days favors a shift of vegetation type towards more southern species, i.e., transformation of tundra environment into shrubs and forest vegetation. Shrubs and trees decrease surface albedo and have a warming effect, especially in winter, but the higher amount of biomass will increase the terrestrial carbon sink.'

*L34: rephrase to 'that biotic factors interact with abiotic climatic factors...'*

Changed as recommended.

*L35: what is meant by 'major species'? dominant species?*

Yes, changed to dominanat species.

*L 38: remove 'really'*

Changed as recommended.

*L67: rephrase to 'the sites are covered by historical high-resolution satellite imagery (Corona archive)'*

Changed as recommended (L 66).

*L68: what is meant by 'projective'? remove this term here*

'Projective forest cover' means surface area covered by forest relative to the total surface area. We removed this term as recommended (L 67).

*L79: 'in flat and hilly terrain' where is which vegetation type dominant? rephrase*

Both vegetation types are represented in both areas: actually it says 'in flat or hilly terrain'.

*L91 - entire section 2.1.2 move to vegetation dynamics - this section here is not well placed and interrupts the flow in the methods. These measurements are only used in the discussion, and sampling is very low (i.e. only 3 data points for e.g. ALT measurements in one place is really not enough to cover the spatial heterogeneity!), so maybe could be shifted to appendix.*

We moved the section to Appendix B.

*L 93 - term 'background area' - either define when first used (area that was never burned?) or use 'control area' throughout the manuscript*

We changed to 'non-disturbed tundra' or 'non-disturbed area' throughout the manuscript (defined in L54, re-iterated in L 164).

*L123: Introduce Corona imagery with a short description and reference.*

We added the following sentences (L116-119): 'Corona is the US program from 1958-1972, which used satellite surveillance systems to get high-resolution photographic coverage from USSR-China and some other territories (Ruffner, 1995). The ground resolution of the imagery for subsequent KH-1 to KH-4 missions continuously improved from ca 12 m (KH-1, 1960) to ca 1.5 m (KH-4, 1967).'

Ruffner, K. C. (Ed.) (1995). Corona: America's first satellite program. History Staff, Center for the Study of Intelligence, Central Intelligence Agency.

*L128: replace 'before' with 'at the start of' and remove 'started' at end of sentence*

Changed as recommended (L 123-124).

*L129: indicate data portal used for Landsat data, reference, and what product was downloaded (surface reflectance? if DN, was data calibrated?)*

The data portal for Landsat data is referenced under Table 3 (similar to other data, e.g. Table 4) and under Table 5. We used Level 1 data for fire analysis (added in L 125). All the data were DN. The data used for fire analysis was not calibrated because geometric correction is sufficient for the calculation of area (Song et al., Classification and change detection using Lansat TM data: when and how to correct atmospheric effects? Remote sensing of environment (2001) 75:230-244). We used Level 2 data (surface reflectance accounted, corrected and calibrated) for NDVI analysis (L 160).

*L141: 'when necessary' - what was criteria here?*

Added in L 140-141: The segments with 40-90% of pixels, located at the perifery of large fires, were added to burned area based on visual imagery check.

*L151: time steps between fire and imagery differ a lot through time! what are the consequences on the vegetation succession?*

The time steps ranged from 11-12 years in 1960-2000 to 2-4 years after 2010. The recovery of vegetation is a relatively slow process and burned areas can be reliably identified 15-20 years after the fire. An example for fire in forest-tundra ecotone in 1990 is given below.

[Figure]

*L154: Section on vegetation dynamics: introduce change classes (original status -> status after transition, e.g. tundra -> woodlands). This will also help to improve understanding of related results figure!*

We added the following sentence (L 189-191): 'We introduced three change classes: 'no change', when tundra was identified both in the old and modern images; 'to woodlands', when tundra turned to woodlands; and 'to forests' when tundra turned to forests.'

*L157: rephrase 'maps to satellite data of high spatial resolution'*

Changed as recommended (L 153-154).

*Lines 155-179: This entire section is very confusion as it does not follow a clear logic - reorder by variable and describe data set, processing, and resulting variable one after the other (now all is repeated and mixed up in these 3 sections).*

The section was rewritten as follows (L 173-191):

The visual method can be described as follows. The data sets used in the analysis of vegetation dynamics are summarized in Table 5. The initial state of vegetation was assessed using Corona mosaic (Section 2.3) and topographic maps. The topographic maps were used to develop forest and drytundra masks using automatic tracing in software EasyTrace (v. 8.7). The resulting vector layer was checked and correctedusing Corona mosaic.

Assessment of the current state of vegetation was based on Resurs-P and SPOT data (Table 5). The study areas were almostfully covered by the mosaic of SPOT-6,7 imagery. Ca 10% of the study areas were covered by three paths of Resurs-P (product level 2A, channels B, G, R and NIR). Resurs-P data was co-registered to the SPOT mosaic in ArcGIS. The data were synthesized in the visible radiation range without color adjustment. The SPOT mosaic was used as a pluggable webmap layer withoutadditional processing.

Firstly, based on the topographic map and Corona mosaic, we identified two types of dry tundra sites: those burned before1968 (Sec. 2.3) and those not affected by fires during the whole study period. The area of burned tundra was 1090 km$^2$ and the area of background tundra was 5300 km$^2$. Then we introduced random sample circles with 100 m diameter within the subsets.The samples were spaced at intervals of 2 km within burned tundra (157 samples) and at intervals of 5 km within backgroundtundra (231 sample). We included only the circles, for which we could confidently state that there were no trees in 1968. In the modern imagery, we distinguished between the following types of vegetation: tundra (no trees), woodland (trees coveredless than 50% of the area inside the circle) and forest (trees covered more than 50%of the area inside the circle). We visually compared the areas inside the circles in the historical and modern images (see examples in Fig. 3). We introduced three change classes: 'no change', when tundra was identified both in the old and modern images, 'to woodland', when tundra turned to woodland and 'to forest' when tundra turned to forest.

*L181: NDVI represents vegetation 'greeness', not greening (greening only if positive trend of NDVI over time*

We agree and changed as recommended (L 156).

*L190: replace 'flight' by 'overpass' - but this sentence needs rephrasing in general*

We have split the sentence in two (L 164-165): 'We considered the territories covered by one satellite overpass. Moreover, analysis of burned tundra included only the sites burned once within 60 years.'

*L204: burned tundra?*

Yes, we changed as suggested (L 199).

*L209-211: introducing this section with temperature descriptions >0° is not useful. I advise to start with the general annual mean and monthly trends and then only refer to >5°C as this is what is addressed in this article (i.e. growth-related information).*

We agree with the comment and changed as suggested. However, we want to retain the information about minimum winter temperatures because some bioclimatic models (e.g. diagram in Woodward, 2004) use minimum winter temperatures as a criterion for trees survival. Now it reads (L 204-209):

'During the last 50 years, the mean annual temperature at the meteorological stations has increased by 2.6-3.0°C (Fig. 4): from -9.5°C to -6.5°C in the north and from -6.5°C to -4°C in the south. In addition, the daily minimum temperature has increased by 4-5°C (from -44…-45°C to -40°C, SM, Fig. SA1-SA3).

The growing season includes June-September in the north and May-September in the south (SM, SA4). The length of the growing season (SM, Fig. SA1-SA3), has increased by 24 days: from 72 to 97 days in the north (Novy Port) and from 97 to 121 days in the south (Nadym).'

*L213: what is meant by '...' for temperature indications?*

This means interval of negative temperatures to avoid double '-' symbol.

*L217: add 'temperature of the growing season'*

Changed as recommended (L 209).

*L235: what is meant by 'modelled'? expected based on table 5 classification? Projected by xxx?*

Expected based on current Table 2 classification. We removed the word 'modelled' to avoid confusion.

*L235-241: results are more accessible if in tabular format, i.e. where is vegetation expected to change based on table 5 and retrieved climate indices.*

We added the following table:

Table 6. Change in vegetation class at meteorological stations in accordance with Table 2 (after Tchebakova et al., 1994).

| Meteorological station | Vegetation class, 1960s | Vegetation class, 2010s |
|---|---|---|
| Nadym | Dark-needled northern taiga | Dark-needled middle taiga |
| Nyda | Dark-needled northern taiga | Dark-needled northern taiga |
| Novy Port | Spruce-larch forest-tundra | Dark-needled northern taiga |

*L248: 30% - how does this number relate to Fig 7b results?*

We have added 20 and 30 in the axis labels (Fig. 6b).

*L249: remove 'Table 5'*

Changed as recommended (L 239).

*L274-L276: these 2 sentences are more appropriate for the discussion section as they are an interpretation of results.*

We moved the sentence about lichen to Discussion (L 352) and deleted the sentence about forest (because there was already a similar one in Discussion).

*L295: unclear what is meant here: largest part of burned tundra was in taiga zone?*

Here tundra is type of vegetation, see the answer to major comment.

*L370-378: move this section to the discussion parts on fire.*

Changed as recommended, this section is now at L 334-341.

*L380: this is not a good fit to start the conclusions - this has not been discussed before and is not referenced here.*

We removed the first sentence, which remained from the old version of the manuscript.

*Fig 7b - change axis labelling*

We have changed y-axis labelling. Now number 30, referred in the text, is added in the labels (see also the comment to *L248*). We added grid lines which makes it easier to see the fraction of burned area for different periods (Fig. 6b).

*Fig 8b: convert y-axis and values to area (instead of pixels)*

Changed as suggested (Fig 7b).

*Fig 11: what are the change classes here in this figure? introduce definitions of vegetation stages and transitions in methods and also clarify in this figure.*

We added the definition of change classes in Methods, Sec. Vegetation dynamics (L 189-191) and in the caption to Fig. 11 (current Fig. 10). 'We introduced the following change classes: 'no change', when tundra was identified both in the old and modern images, 'to woodland', when tundra turned to woodlands and 'to forest' when tundra turned to forest.'

*Table1: sort vegetation distribution cell contents along N-S vegetation types or percentages*

We sorted cell contents by percentages.

*Table 2: move this table to appendix, as data are only used in discussion. Also, number of ALT measurements is too low to be representative (e.g. n=3 and n=5); this does not seem to be a full species list, but only dominant species? Also, if species level determination is not available, use sp (e.g. Ledum sp., Polytrichum sp.)*

We moved the table to Appendix B.

Number of ALT measurements has been determined in accordance with the guide book 'Field geocryological studies' ed. by I. Ya. Baranov, Moscow, 1961 [in Russian]. We understand, however, that this is an old literature source and the standards have changed. We will take this into account performing new measurements.

The table reports the dominant species (added). We added level determination: *Ledum palustre, Politrichum commune.*

*Table 4: reference for USGS is a sentinel website?*

Changed to earthexplorer.usgs.gov

*Table 7: move to appendix*

The table is moved to Supplementary material (Table S1).

We thank again the referee for the useful suggestions. We hope that the manuscript is now suitable for publication in BG.

[revised manuscript text omitted]

---

## Author Response (AR3)

**Replies to Editor's comments**

*1. In the definition of vegetation zones and types, please state clearly when a woodland and forest category falls into one of the taiga zones, because you have defined several subzones of taiga. Adding table 6 is certainly helpful. Please make sure this re-classification is clearly described in the methods and consistently used throughout the manuscript. When you use the term "forest", please specify which type of taiga it now is, being consistent with table 6.*

We apologize for not being clear enough. Adding table 6 possibly put too much accent on the classes defined by the bioclimatic model whereas we mostly used the classes following zonation of the topographic map by Ilyina et al. (1985) throughout the manuscript (e.g., Fig. 1, Fig. 9, Fig. 10). Ilyina et al. do not introduce sub-zones of taiga. The map is developed based on surveys and thus, it is more precise than models. Moreover, different bioclimatic models give different predictions. For example, using the diagram from Woodward et al., Global climate and the distribution of plant biomes, Phil. Trans. R. Soc. Lond. B 2004 359, 1465-1476 (their Fig. 1), one can state that in 1960s, our study areas were within tundra zone and in 2010s, only the areas near Nadym turned into boreal forest zone. Thus, both bioclimatic models predict change of zones but the boundaries of these zones are different. We used SiBCliM because we expected it to be more precise in Siberia. However, the model predictions of zones do not agree with topographic maps.

We added the following sentences in Discussion (L310-312):
'Note that the boundaries of vegetation classes as defined by the topographic maps do not coincide with those of the classes based on SiBCliM (Table 5). This is likely the consequence of the insufficient precision of the bioclimatic model, which can be too rough in transitional areas.'

In Table 6, we added a column with the zones based on topographic map (Ilyina et al., 1985). In Methods, we added the following sentence (L79-80): 'Further on, we use the definition and boundaries of vegetation zones following topographic maps.'

*2. Your response to*
*L213: what is meant by '...' for temperature indications?*
*your reply: This means interval of negative temperatures to avoid double '-'symbol.*
*Please change to "from ... to...".*

We have changed 'from -44…-45°C to -40°C' to 'from -44 or -45°C to -40°C'

*3. Your response to*
*L295: unclear what is meant here: largest part of burned tundra was in taiga zone?*
*your reply: Here tundra is type of vegetation, see the answer to major comment.*
*Please rephrase to make clear which vegetation type was affected at the time the fire occurred and what is the vegetation zone now (I assume taiga, but which sub-type?).*

We follow the same definition of vegetation zones throughout the paper, it is based on topographic maps (added the sentence in Methods). The main purpose for introducing classes as defined by the bioclimatic model is to show that the whole study area is now theoretically suitable for forest. This is also stated in results, L232.

This is how this part of the sentence looks in the context: 'Fig. 9 illustrates the vegetation shift from dry tundra to other types of vegetation in the sites burned before 1968. The largest part of burned dry tundra was in the northern taiga zone, and relatively large part was in the forest-tundra ecotone.'

We are grateful to the Editor for the comments, which helped to improve the clarity of the manuscript.

[revised manuscript text omitted]